# A Study in Scarlet: Integrative Taxonomy of the Spider Genus *Loureedia* (Araneae: Eresidae)

Tamás Szűts [1,*], Krisztián Szabó [1,*], Alireza Zamani [2], Martin Forman [3], Jeremy Miller [4], Pierre Oger [5], Magali Fabregat [6], Gábor Kovács [7] and János Gál [8]

1    Molecular Ecology Research Group, Department of Ecology, University of Veterinary Medicine Budapest, Rottenbiller utca 50, 1071 Budapest, Hungary
2    Zoological Museum, Biodiversity Unit, University of Turku, 20014 Turku, Finland
3    Laboratory of Arachnid Cytogenetics, Department of Genetics and Microbiology, Faculty of Science, Charles University, Viničná 5, 128 44 Prague, Czech Republic
4    Understanding Evolution Research Group, Naturalis Biodiversity Center, Darwinweg 2, 2333 CR Leiden, The Netherlands
5    Rue du Grand Vivier 14, 4217 Waret l'Evêque, Belgium
6    Association Française Arachnologie, Rue Buffon 61, 75005 Paris, France
7    Londoni Krt. 1., 6724 Szeged, Hungary
8    Department of Exotic Animal and Wildlife Medicine, University of Veterinary Medicine Budapest, István u. 2, 1078 Budapest, Hungary
*    Correspondence: szuts.tamas@univet.hu (T.Sz.); szabo.krisztian@univet.hu (K.Sz.)

**Abstract:** The eresid spider genus *Loureedia* (Miller et al., 2012) was described a decade ago, despite its type species being described in the mid-19th century, which illuminates the difficulties in obtaining specimens. The genus was initially described as monotypic. Ever since, four other species have been assigned to *Loureedia*, including three newly discovered ones. Primarily due to the extravagant appearance of the males, stories about the discovery of species of *Loureedia* have been the subject of relatively wide media coverage over the years, leading to numerous new populations and putative undescribed species being documented by naturalists and citizen scientists. These species, although bearing distinct differences in their coloration patterns, typically vary only slightly in the structure of their copulatory organs, the primary traits used in spider systematics. This highlights an important taxonomic problem: while it is easy to diagnose the genus or recognize the species that belong to it, it is challenging to differentiate the species from one another, particularly when using only a single line of evidence. In this paper, we have tackled this issue using an integrative approach, i.e., a combination of molecular markers (the mitochondrial COI) and traditional morphological characters. The effects of different observational angles on the perceived shape of the conductor are discussed. Except for one species, we obtained DNA data of all members of the genus. Based on these data, the first phylogeny for *Loureedia* is presented, and two North African species, *Loureedia maroccana* (Gál et al., 2017) and *Loureedia jerbae* (El-Hennawy, 2005), are revalidated from synonymy. The distribution records of all described species are mapped.

**Keywords:** ASAP; DNA barcode; Joker spider; Middle East; North Africa; phylogeny

## 1. Introduction

Despite its recent and detailed description, a concise diagnosis [1], and the strikingly distinct appearance of its species (Figure 1), the taxonomy of the velvet spider genus *Loureedia* Miller et al., 2012 (Eresidae) remains challenging at the species level. Specimens are still rare in collections, and single-species descriptions lack cohesive comparative diagnoses that effectively discriminate among the species or sufficient anatomical details, particularly showing all male palpal characters [2]. Before the establishment of the genus, some *Loureedia* species were described in *Eresus* Walckenaer, 1805 [1,3,4]. Moreover, preserved specimens can bleach out quickly, lose the coloration of their setae, and lose their colored scales, which

further complicates efforts to match descriptions to living or freshly collected specimens. In their comprehensive atlas of the velvet spiders, Miller et al. [1] proposed *Loureedia* to accommodate a single species, *L. annulipes* (Lucas, 1857), described from the "environs of Rio de Janeiro" (surroundings of Rio de Janeiro) [4]. However, this was found to be an error by Lucas [4]; a re-examination of the label of the vial containing the holotype male (AR5391) reads *Patria Ignota* (unknown locality) [1]. In fact, only one neotropical eresid species has been recorded so far: *Stegodyphus manaus* Kraus and Kraus, 1992 [5].

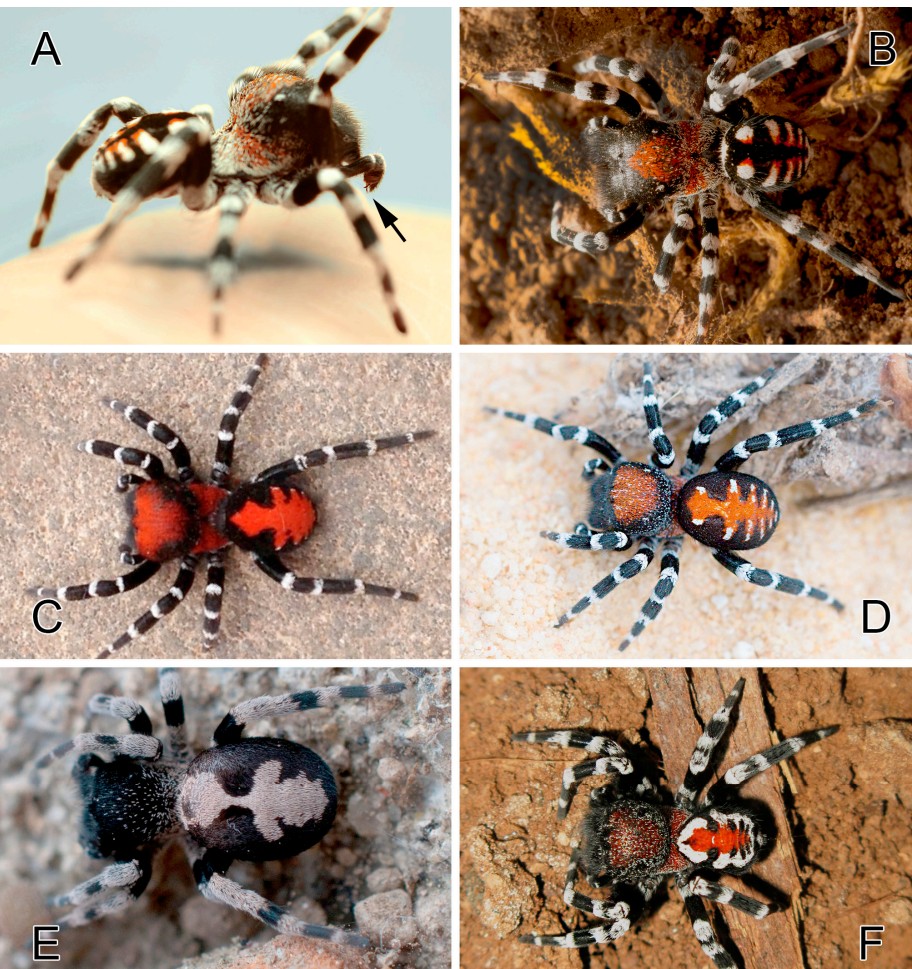

**Figure 1.** Males of five species of *Loureedia*. (**A**) *L. annulipes* from Israel, in a defensive posture, with the bifid conductor visible (arrow) (photo: Tamás Szűts); (**B**) *L. annulipes* from Israel (photo: Tamás Szűts); (**C**) *L. maroccana* from Morocco (photo: János Gál); (**D**) *L. jerbae* from Djerba, Tunisia (photo: Stanislav Macík); (**E**) *L. colleni* from Spain (photo: Magali Fabregat); (**F**) *L. phoenixi* from Iran (photo: Alireza Zamani). (**C**) reproduced from Gál et al. [2]; (**F**) reproduced from Zamani and Marusik [6].

Furthermore, Miller et al. [1] considered *Eresus semicanus* Simon, 1908 and *Eresus jerbae* El-Hennawy, 2005 to be junior synonyms of *L. annulipes*, which at that time was assumed to be distributed in Israel, Egypt, and Tunisia.

Since the publication of Miller et al. [1], three additional species have been described in this genus: *Loureedia maroccana* Gál et al., 2017 from Morocco [2], *Loureedia colleni* Henriques, Miñano and Pérez-Zarcos, 2018 from Spain [3], and *Loureedia phoenixi* Zamani and Marusik, 2020 from Iran [6]. These new taxonomic findings significantly expanded the known range of the genus, hinting at a higher diversity in the group than previously assumed. This became further evident in several reports and observations of populations in North Africa and the Middle East, some of which did not seem to belong to any of the known species [7]. Finally, Henriques et al. [3] proposed *Loureedia lucasi* (Simon, 1873) from Algeria as a new combination (ex. *Eresus*) and considered the recently described *L. maroccana* to be its junior

synonym, resulting in a total number of four valid species currently considered in the genus (Figure 2).

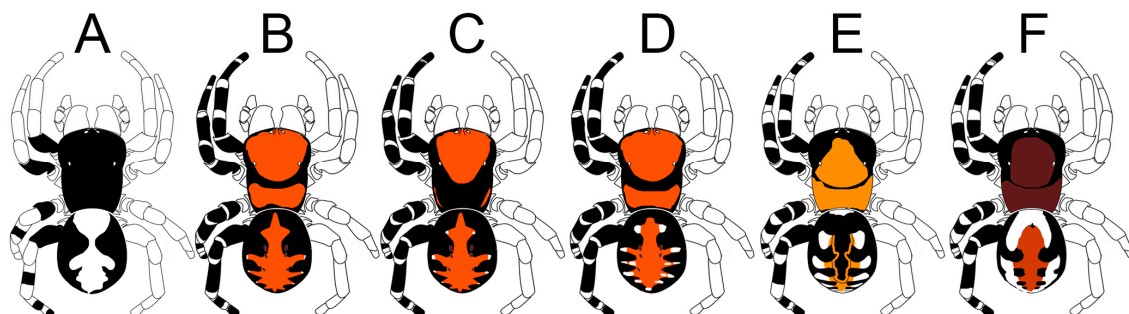

**Figure 2.** Male coloration patterns of six species of *Loureedia*. (**A**) *L. colleni*; (**B**) *L. maroccana*; (**C**) *L. lucasi*; (**D**) *L. jerbae*; (**E**) *L. annulipes*; (**F**) *L. phoenixi*.

Although the establishment of *Loureedia* is recent, its type species (*Eresus annulipes*) was described over a century ago [4]. The original description [4], on the basis of a single male specimen, was rather detailed. Regardless, it mentions the most conspicuous character to delimit the genus: "*l'organe excitateur est rougeàtre, globuliforme, et armé à son extrémité d'un crochet corné, noir, bi-épineux.*" [The bulb is reddish, globuliform, and armed at its extremity with a horny, black, bi-spinous hook.].

In his description of *Eresus lucasi*, Simon [8] did not mention any bifid palpal structure, despite providing a rather detailed description. Although we were unable to examine the holotype of *E. lucasi*, the available online digital images [3] allowed us to compare it with other specimens. Subsequently, in the description of *Eresus semicanus*, Simon wrote [9] "*le processus du bulbe inégalement bifide.*" [the process of the bulb is unequally bifid], thus proving that he was indeed aware of that feature. For this species, diagnosable drawings and a translation of the original description were provided by El-Hennawy [10]. El-Hennawy later described another species, *E. jerbae* El-Hennawy, 2005 [11], from the island Djerba (Tunisia).

Considering that the known species diversity and geographical range of the genus has noticeably expanded over the past decade, we felt compelled to summarize the current state of knowledge regarding the taxonomy and diversity of the group, explore its species delimitation with molecular data, and provide a preliminary phylogeny and a comprehensive overview of their most relevant characters.

Herein, we redescribe and illustrate *L. maroccana* based on the type series and non-type material and reject its synonymy with *L. lucasi*. Moreover, we reinstate *L. jerbae* on the basis of morphological and molecular data obtained from fresh material collected in its type locality and describe its male for the first time. Finally, we provide a description of the unique retreat structure of *L. colleni* and map the distribution records published so far of the representative of the genus.

## 2. Materials and Methods

### 2.1. Morphology

Morphological studies were carried out using a Nikon S800 dissection microscope and a Nikon Eclipse E200 compound microscope. Digital images were obtained using a Nikon D300S DLSR attached to the former and a Tucsen TrueChrome Metrics attached to the latter.

Specimens were photographed, using the dissection microscope, in a Petri dish filled with sand. Palps were photographed, using the compound microscope, in a small Petri dish filled with hand sanitizer (for precise angle manipulation) and covered with a coverslip. SEM images were taken with a Hitachi S-4700 microscope at the Department of Applied and Environmental Chemistry, University of Szeged, Hungary. The lengths of leg segments

were measured on the dorsal side and are listed as the total length (femur + patella + tibia + metatarsus + tarsus). All measurements are in millimeters.

### 2.2. Terminology

The terminology mostly follows Miller et al. [1], with slight modifications. For the conductor (Figure 3), the terminology of Henriques et al. [3] and Zamani and Marusik [6] is used. New terms are herein applied to the lamellar parts of the pro- and retrolateral arms of the conductor (Figure 3; pal and ral, respectively). Pal is a new term for the *flame-shaped lamellated structure* described by Henriques et al. [3], as it has a more precise descriptive power. There is slight confusion regarding what Henriques et al. [3] meant by frontal margin. They referred to it as being serrated in *L. maroccana*, where the frontal margin is in fact smooth, but both the pal and ral are serrated. The various terms are summarized in Figure 3.

### 2.3. Abbreviations

The following abbreviation were used: ALE—anterior lateral eye; AME—anterior median eye; bcm—basal conductor margin; ecm—ectal conductor margin; em—embolus; fcm—frontal conductor margin; mcm—mesal conductor margin; pa—prolateral arm of the conductor; pal—lamella of prolateral arm; PLE—posterior lateral eye; PME—posterior median eye; ra—retrolateral arm of the conductor; ral—lamella of retrolateral arm; sc—shoulder of the conductor; st—stem of the conductor.

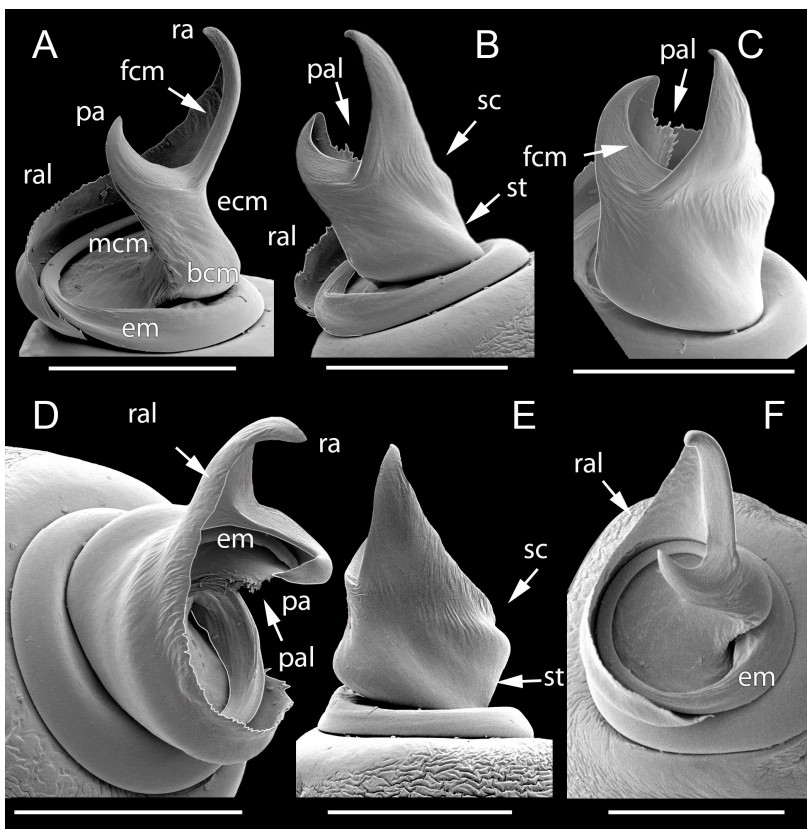

**Figure 3.** SEM images of the conductor and embolus of *Loureedia maroccana*, male paratype. (**A**) oblique ventral view; (**B**) oblique retrolateral view; (**C**) apico-retrolateral view; (**D**) apico-prolateral view; (**E**) retrolateral view; (**F**) apico-ventral view. Abbreviations: bcm: basal conductor margin, ecm: ectal conductor margin, em: embolus, fcm: frontal conductor margin, mcm: mesal conductor margin, pa: prolateral arm of the conductor, pal: lamella of prolateral arm, ra: retrolateral arm of the conductor, ral: lamella of retrolateral arm, sc: shoulder of the conductor, st: stem of the conductor. Scale bars: 0.25 mm.

### 2.4. Molecular Methods

Tissue samples were obtained from freshly collected specimens preserved in 96% or 70% ethanol. Whole genomic DNA was extracted from one or two separated legs using standard extraction kits. The barcoding segment of the mitochondrial cytochrome oxidase I (COI) using the LCO1940/HCO2198 primers [12] was amplified using PCR, following the publisher's protocols. Sequences were edited using Gap4 of the Staden Package [13] and were deposited in GenBank; the accession numbers are reported in Table 1.

**Table 1.** Voucher data and sequence IDs.

| Species | Specimen ID | Locality | Seq ID |
| --- | --- | --- | --- |
| *Adonea fimbriata* | SAJ61 | Israel | OL352214 |
| *Eresus crassitibialis* | SAJ27 | Canary Islands | OL352233 |
| *Eresus hermani* | SAJ123 | Hungary | OL352225 |
| *Eresus kollari* | SAJ119 | Hungary | OL352223 |
| *Eresus moravicus* | SAJ100 | Hungary | OL352219 |
| *Eresus walckenaeri* | SAJ14 | Turkey | OL352229 |
| *Loureedia annulipes* | LEJD1 | Israel | e40320602643 |
| *Loureedia annulipes* | isolate15_10 | Israel | FJ94900 |
| *Loureedia annulipes* | SAJ264 | Israel | OQ307827 |
| *Loureedia colleni* | SAJ02 | Spain | OQ307823 |
| *Loureedia colleni* | SAJ03 | Spain | OQ307824 |
| *Loureedia colleni* | SAJ04 | Spain | OQ307825 |
| *Loureedia jerbae* | SAJ262 | Djerba, Tunisia | OQ307829 |
| *Loureedia maroccana* | SAJ36 * | Morocco | OQ307826 |
| *Loureedia maroccana* | LIV | Morocco | KX443583 |
| *Loureedia phoenixi* | LEJD2 | Iran | e4032060267 |
| *Stegodyphus dufouri* | SAJ34 | Egypt | OQ307830 |

* Holotype.

### 2.5. Molecular Analyses

We used Mega 11 [14] with default settings to align the sequences. Some previously published sequences were aligned and analyzed together with our own samples [1,2,15].

The phylogenetic relationships of *Loureedia* within the clade consisting of *Eresus*, *Adonea* Simon, 1873, *Loureedia*, and *Dorceus* C. L. Koch, 1846 with *Stegodyphus* Simon, 1873, as sister to them, is well established [1]. Hence, the following outgroups were used (Table 1): *Stegodyphus dufouri* (Audouin, 1826); *Adonea fimbriata* Simon, 1873; *Eresus walckenaeri* Brullé, 1832; *Eresus kollari* Rossi, 1846; *Eresus moravicus* Řezáč, 2008; *Eresus hermani* Kovács et al., 2015; and *Eresus crassitibialis* Wunderlich, 1987.

The same taxa were used as a reference in the assemble species by automatic partitioning (ASAP) analysis [16]. The phylogenetic tree was inferred via Bayesian inference (BI) in MrBayes v. 3.2.6 [17] using the GTR+I+G model of sequence evolution, based on the results of a jModelTest2 analysis [18]. The MrBayes analyses consisted of two independent runs with one cold chain and five hot chains from random starting trees, which were run for 10 million generations and sampled every 1000 generations. Convergence was assessed through an examination of the standard deviation of split frequencies, which was well below the recommended threshold (0.01). We also used Tracer 1.7 [19] to evaluate convergence and to verify that effective sample sizes were >200 for all parameters. The two MrBayes runs were combined after the deletion of burn-in generations (25%), and a majority-rule consensus phylogram was created. The resulting phylogram was edited and visualized with FigTree v.1.4 [20].

Based on the COI sequences, species delimitation was also attempted via a genetic-distance-based automatic partitioning method using the ASAP software [16]. The ASAP software web interface was used, K2P was selected as the nucleotide substitution model, and the other parameters were left with default settings. The ASAP delimitation was defined by evaluating both partitions with the first- and second-best ASAP scores.

### 2.6. Specimen Depositories

The specimens were deposited at the following institutes: HNHM—Hungarian Natural History Museum (Eszter Lazányi-Bacsó); HUJ—Hebrew University of Jerusalem, Jerusalem (Efrat Gavish-Regev); MHNG—Muséum d'histoire naturelle de la Ville de Genève (Lionel Monod); NMHN—Muséum national d'Histoire naturelle, Paris (Christine Rollard); PCJG—Private Collection of János Gál; ZMUT—Zoological Museum of the University of Turku (Varpu Vahtera).

### 2.7. Distribution Data and Map

Distribution data were downloaded from GBIF [21], and only published data were used [3,6] (i.e., iNaturalist data were not used, unless certain ID was possible). Our voucher *Loureedia* specimens, with additional data from Iran, were also supplemented [6,22]. The distribution map was produced by SimpleMappr [23].

### 3. Results

*Taxonomy*

*Loureedia* Miller, Griswold, Scharff, Řezáč, Szűts and Marhabaie, 2012.
*Loureedia* Miller et al., 2012: 81 [1] (original description).
*Loureedia*: Henriques et al. 2018: 5 [3]; Zamani and Marusik 2020: 239 [6].
Type species: *Eresus annulipes* Lucas, 1857, *Patria ignota* (unknown site).
Diagnosis. The most diagnostic character of the genus is the bifid conductor of the male palp (Figure 4). See also Miller et al. [1].

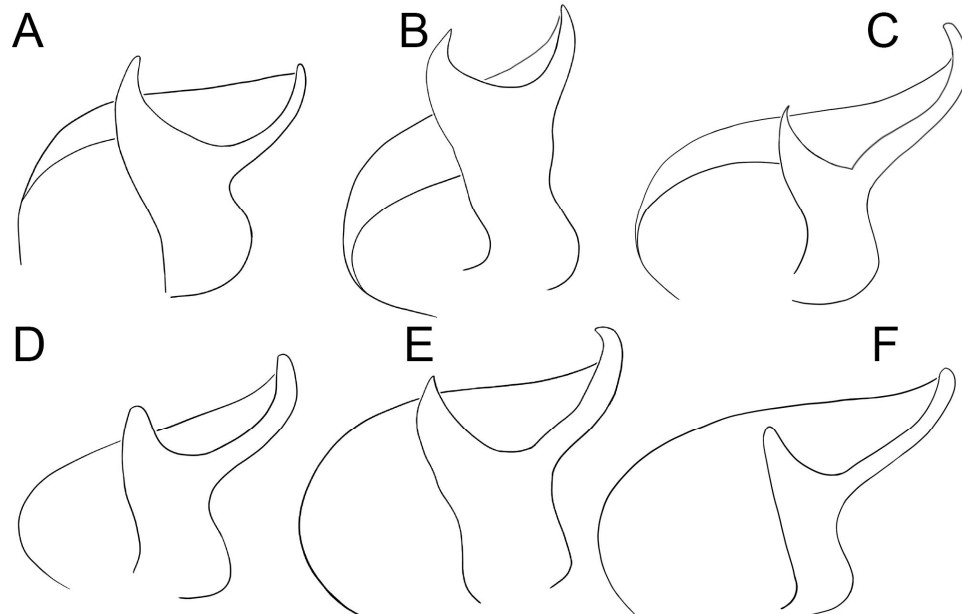

**Figure 4.** Conductors of six species of *Loureedia* (ventral view). (**A**) *L. phoenixi*; (**B**) *L. jerbae*; (**C**) *L. annulipes*; (**D**) *L. lucasi*; (**E**) *L. maroccana*; (**F**) *L. colleni*. A, D, and F are based on Zamani and Marusik [6]. Line drawings by Mahla Pourcheraghi.

Composition. Six species: *L. annulipes*, *L. colleni*, *L. jerbae*, *L. lucasi*, *L. maroccana*, and *L. phoenixi*.

Distribution. Spain, North Africa, and the Middle East (east to Iran) (see Figure 5).

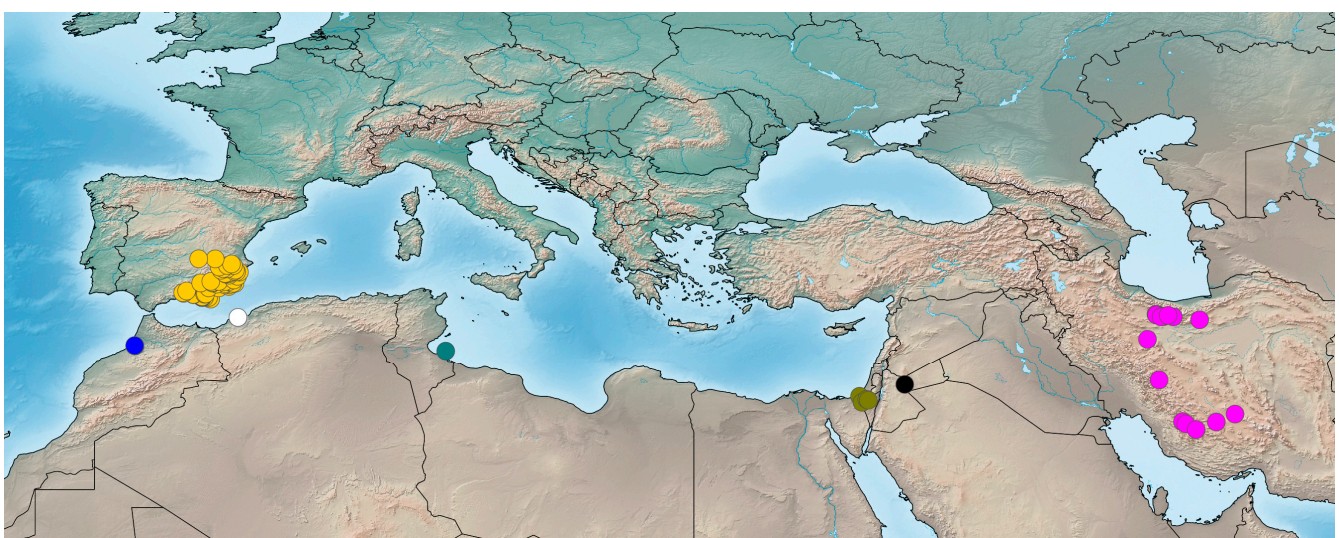

**Figure 5.** Distribution records of species of *Loureedia*. Color codes: yellow—*L. colleni*, blue—*L. maroccana*, white—*L. lucasi*, green—*L. jerbae*, khaki—*L. annulipes* (SAJ264 not shown), magenta—*L. phoenixi*, black dot—unidentified *Loureedia* specimen from Jordan [7].

*Loureedia annulipes* (Lucas, 1857).

Figure 1A,B, Figures 2E, 4C, 5, 6C, 7C and 8C.

*Eresus annulipes* Lucas, 1857: 21 (♂) [4] (original description).

*Eresus semicanus* Simon, 1908: 83 (♂) [9].

*Eresus semicanus*: Simon 1911: 294, Fig 5 (♂) [24]; El-Hennawy 2004: 28, Figs 2A,B, 3A–C and 4A,B (♂♀) [10].

*Stegodyphus annulipes*: Kraus & Kraus 1992: 15, 19 [5].

*Loureedia annulipes*: Miller et al. 2012: 88, Figs 1G–H, 4I, 9I–L, 13G–I, 18A, D, 62A–J, 63A–F, 64A–D, 65A–F, 66A–F, and 67A–F (♂♀) [1]; Henriques et al. 2018: 5, Fig 2a–h (♂♀) [3]; Zamani and Marusik 2020: 242, Fig 3g (♂) [6].

Type material. Holotype: male (AR5391, NMHN), *Patria Ignota* (unknown site) (not examined).

Other examined material. One male (HUJ Ara 16551), ISRAEL: Southern District: Negev desert, 1 km north of Kibbutz Retamim, 29.X.2016 (leg. Reut A. Ein-Gil).

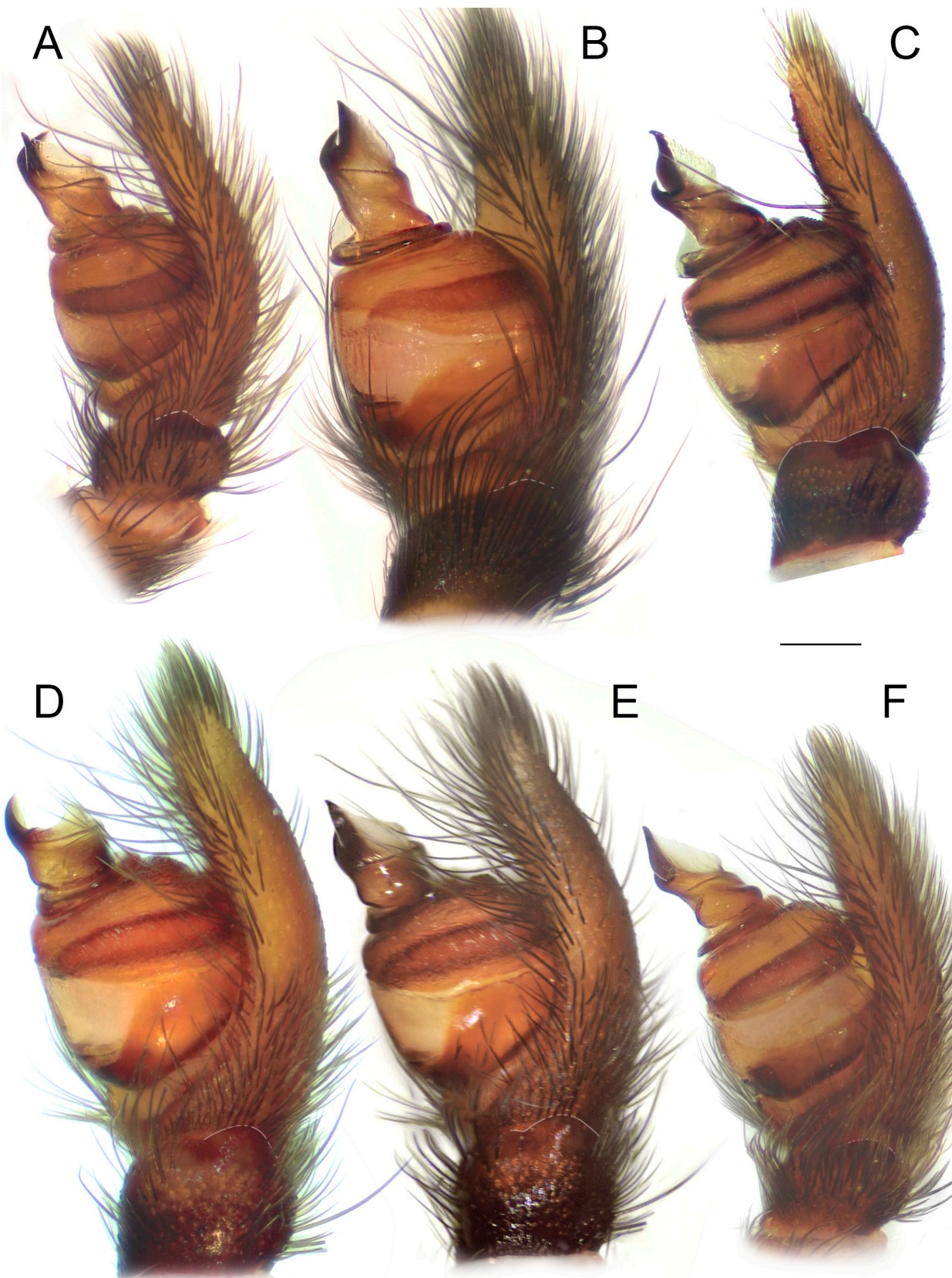

**Figure 6.** Male palps of five species of *Loureedia* (retrolateral view). (**A**) *L. phoenixi*; (**B**) *L. jerbae*; (**C**) *L. annulipes*; (**D**) *L. maroccana*, holotype (after conductor was broken); (**E**) *L. maroccana*, holotype (conductor still intact); (**F**) *L. colleni*. Scale bar: 0.25 mm.

Diagnosis. The male palp of *L. annulipes* (Figures 4C, 6C, 7C and 8C) is most similar to that of *L. colleni* (Figures 4F, 6F, 7F and 8F), with the retrolateral arm of the conductor being much longer than the prolateral arm and bearing a gradual curvature and a blunt tip (see Figure 4C,F). The male of *L. annulipes* can be distinguished from that of *L. colleni* by the wider stem of the conductor (Figure 4C), bearing a distinct concavity on the mesal margin (Figure 4C) vs. a narrower stem (Figure 4F) with an almost straight mesal margin (Figure 4F), the curved tip of the retrolateral arm (Figures 4C, 6C and 7C) of the conductor, and the abdominal coloration pattern consisting of numerous white spots (Figure 1A,B and Figure 2E) and two longitudinal, interrupted yellowish stripes (Figures 1B and 2E) vs. one or two white semi-foliate patterns in some individuals in the form of two large patches (Figures 1E and 2A). The females of the two species can be differentiated by the epigynal fovea (i.e., the median lobe described by Miller et al. [1]), which is almost as long as it is wide in *L. annulipes* (see Miller et al. [1]: Fig 18A) vs. longer than wide in *L. colleni* (see Henriques et al. [3]: Fig 9C).

Description. Male. Habitus as in Figure 1A,B and Figure 2E. Total length: 8.01. Carapace: 4.40 long and 3.65 wide. Abdomen: 3.79 long and 3.20 wide. Eye sizes and inter-eye distances: AME 0.27, PME 0.23, ALE 0.12, PLE 0.12, AME–AME 0.34, and ALE–AME 0.93. The carapace, sternum, labium, chelicerae, and maxillae dark brown. The carapace mostly covered with fine, long, black and shorter white and orange setae. The pars cephalica in most individuals with a localized triangular patch of red scales (absent in some individuals, see Miller et al. [1]: Fig 1G).

The center of the pars cephalica covered with orange setae, and the posterior part covered with fine white setae. Legs covered with thin black hairs, with distinct regions of white hairs at the joints of all segments, forming distinct white annulations. Abdomen velvet black; a foliate pattern with a black median elongated patch forming four pairs of elongated dots with orange markings on the inner parts and white markings on the outer parts. White patches unify at the posterior part of dorsum. Measurements of legs: I: 8.59 (2.99, 1.37, 1.71, 1.57, 0.94); II: 7.91 (2.47, 1.59, 1.41, 1.53, 0.89); III: 6.63 (2.31, 1.10, 1.21, 1.29, 0.70); IV: 9.56 (3.05, 1.82, 2.01, 1.77, 0.89).

Palp as in Figures 4C, 6C, 7C and 8C. The stem of the conductor ca. 1.5 times longer than wide. The mesal and ectal margins of the conductor with slight curvatures. The retrolateral arm of the conductor ca. 2.5 times longer than the prolateral arm, and with blunt tip; prolateral arm with a pointed tip.

Female. Deciphering the identity of females of this species is still in progress. Miller et al. [1] described the females based on both *L. jerbae* and *L. annulipes* specimens. The two females are indeed very similar, and comparative material is still being collected.

Variation. The number of white patches on the dorsal surface of the abdomen varies, typically from four to six pairs. They may be connected to each other at their inner margins in some specimens. There is also variation in the width of the median black stripe and the extent of the orange markings. Some specimens have a white band on the anterior portion of the abdomen.

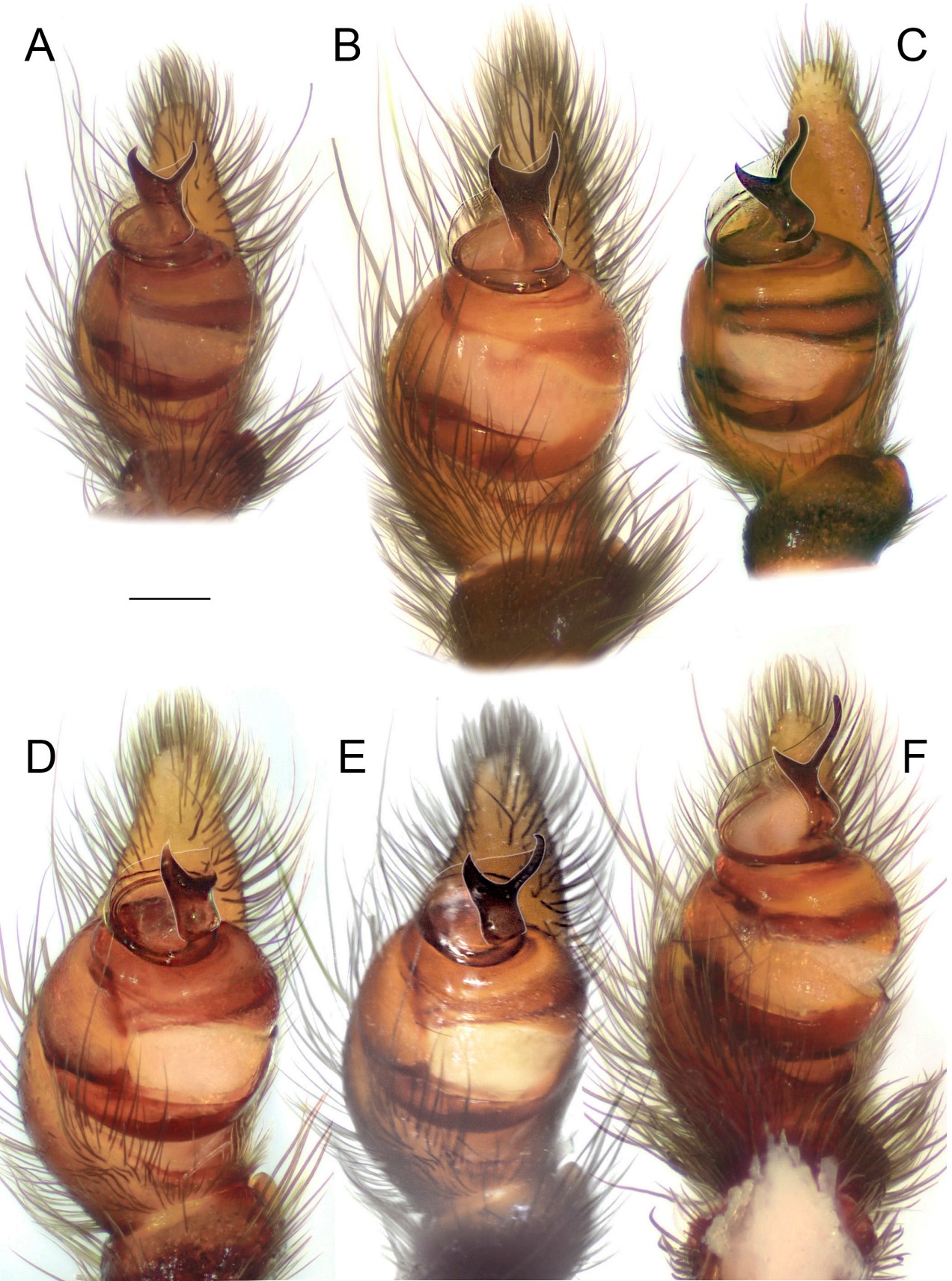

**Figure 7.** Male palps of five species of *Loureedia* (ventral view). (**A**) *Loureedia phoenixi*; (**B**) *L. jerbae*; (**C**) *L. annulipes*; (**D**) *L. maroccana*, holotype (after conductor was broken); (**E**) *L. maroccana*, holotype (conductor still intact); (**F**) *L. colleni*. Scale bar: 0.25 mm.

Natural history. Known from the sandy dunes of the Negev desert (Figure 9A,B).
Phenology. The males are active during October–November.
Distribution. Reliably known only from Israel (Southern District) (see Figure 5).

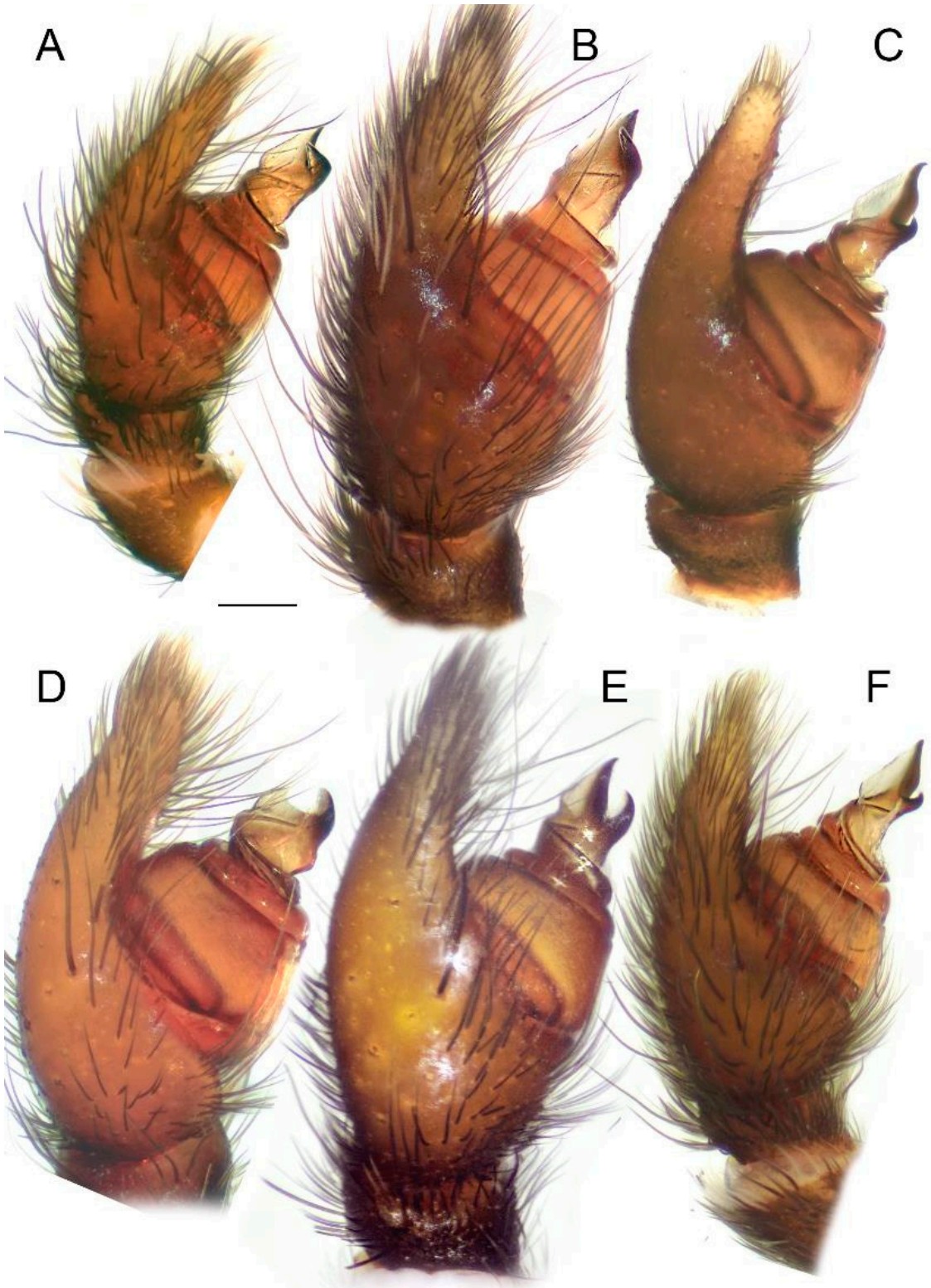

**Figure 8.** Male palps of five species of *Loureedia* (prolateral view). (**A**) *Loureedia phoenixi*; (**B**) *L. jerbae*; (**C**) *L. annulipes*; (**D**) *L. maroccana*, holotype (after conductor was broken); (**E**) *L. maroccana*, holotype (conductor still intact); (**F**) *L. colleni*. Scale bar: 0.25 mm.

*Loureedia colleni* Henriques, Miñano and Pérez-Zarcos, 2018.

Figures 1E, 2A, 4E, 5, 6F, 7F, 8F, 10 and 15.

*Loureedia colleni* Henriques, Miñano and Pérez-Zarcos in Henriques et al., 2018: 8, Figs 3, 4–8a–c, 9a–d, 13b and S–S12 (♂♀) [3] (original description).

*Loureedia colleni*: Zamani and Marusik 2020: 242, Fig 3i (♂) [6].

Type material. Holotype: male (MNCN), SPAIN: Andalucía: Granada province, 820 m a.s.l., 10.X.2010 (leg. Carlos Jerez del Valle) (not examined).

Other examined material. Two males and one female (HNHM 9207, 9209, and 9215), SPAIN: Andalucía: Almería Province, Sierra de Gádor, Vícar, 36°49′03.0″ N, 2°39′14.1″ W, 820 m a.s.l., 10.IX. 2017 (leg. Magali Fabregat).

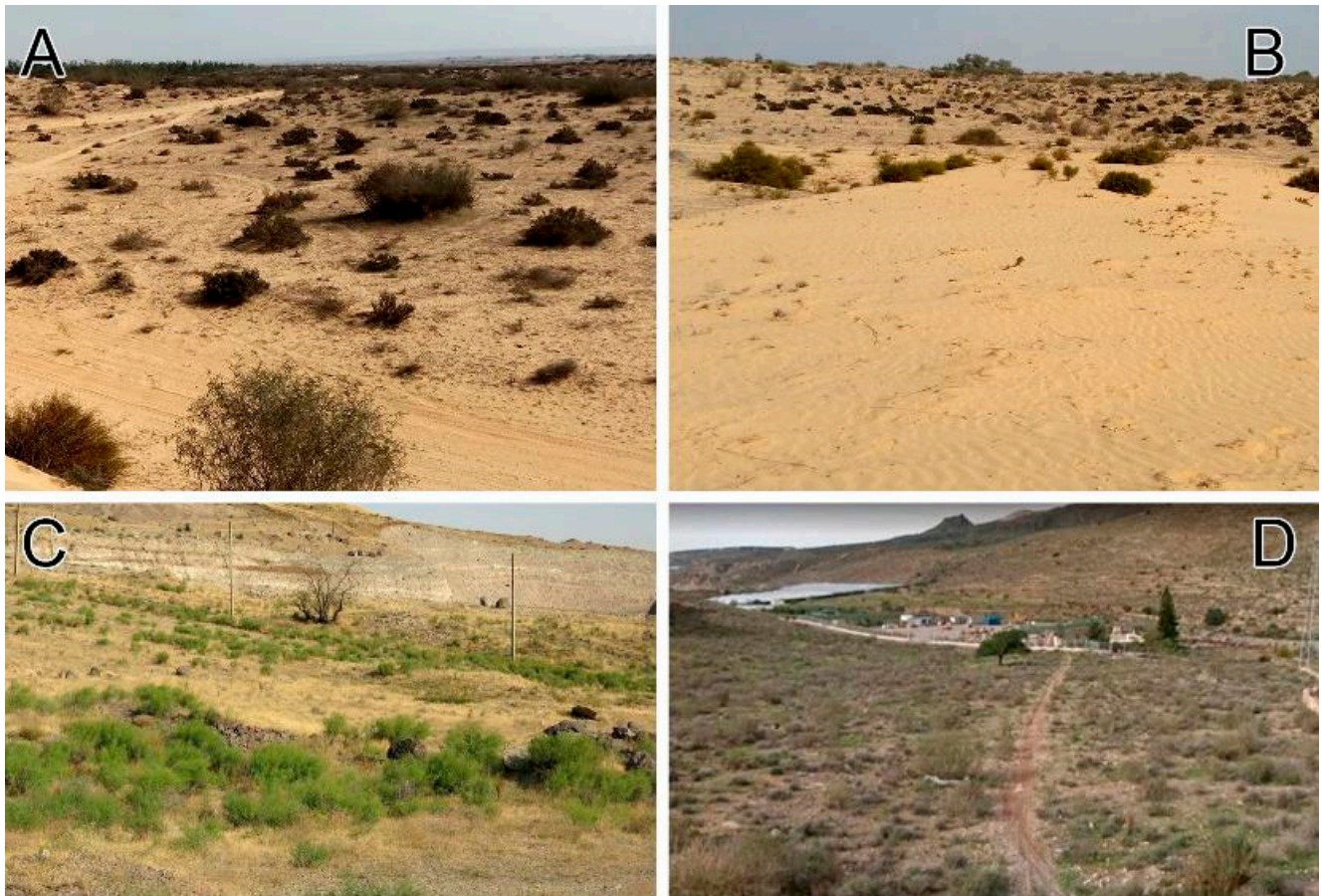

**Figure 9.** Habitats of three species of *Loureedia*. (**A**,**B**) *L. annulipes* from Israel (Reut Ein-Gil); (**C**) *L. phoenixi* from Iran (photo: Alireza Zamani); (**D**) *L. colleni* from Spain (photo: Magali Fabregat).

Diagnosis. This species differs from all of its congeners by the black-and-white coloration pattern of the male (Figures 1E and 2A) vs. having yellowish to scarlet red abdominal patterns (see Figures 1A–D and 2B–F). The male palp of *L. colleni* (Figures 4F, 6F, 7F and 8F) is most similar to that of *L. annulipes* (Figure 4C), as the prolateral arm of the conductor is much shorter than the retrolateral arm (Figure 4F), which bears a gradual curvature (7F). The male of *L. colleni* can be diagnosed by the narrower stem of the conductor (Figure 6F), with an almost straight mesal margin (Figure 4F). The female can be recognized by an epigynal fovea that is longer than it is wide (see Henriques et al. [3]: Fig 8a).

Description. Male. Habitus as in Figures 1E and 2A. Total length: 6.43. Carapace: 3.35 long and 2.84 wide. Abdomen: 3.19 long and 2.55 wide. Eye sizes and inter-eye distances: AME 0.14, PME 0.16, ALE 0.04, PLE 0.08, AME–AME 0.30, and ALE–AME 0.76. The carapace, sternum, labium, chelicerae, and maxillae black. Carapace mostly covered with long black setae and scattered shorter white setae. White setae localized densely on

the pars thoracica and form a triangle on the pars cephalica (Figure 1E). Legs covered with thick white hairs. Abdomen velvet black with a longitudinal median white foliate pattern bearing a distinct mediolateral lobe; the most anterior part of the folium merging and forming a distinct white spot. Measurements of legs: I: 7.15 (2.07, 1.16, 1.45, 1.43, 1.01); II: 6.33 (1.94, 1.19, 1.18, 1.22, 0.78); III: 5.44 (1.85, 0.80, 1.08, 1.01, 0.68); IV: 7.29 (2.32, 1.28, 1.56, 1.39, 0.71).

Palp as in Figures 4F, 6F, 7F and 8F. The stem of the conductor ca. 1.5 times longer than wide. The mesal margin of the conductor almost straight, and the ectal margin with an apical invagination. The retrolateral arm of the conductor ca. 2.5 times longer than the prolateral arm, and both arms with blunt tips.

Female. See Henriques et al. [3].

Variation. A wide array of abdominal pattern variations has already been illustrated [3]. The highest amount of variation occurs in the white foliate pattern, which may either be solid or form two large separate patches. Here, we examined two distinct color pattern forms (Figure 15A,B). Minor variations also occur on the male palp; these are considered intraspecific variations, as the COI sequences of the two males were identical, whereas they were slightly different (99.965% similarity) from that of the female.

Natural history. The species' habitat preference has already been described [3]. The examined specimens were collected on a hillside with south and south-east exposure in a semi-arid open area (Figure 9D) with short, sparse vegetation. The vegetation in this area mainly consists of degraded bushes, tufts of thyme, thorny broom, *Launaea arborescens*, and *Ononis natrix hispanica*. The soil is mainly puddingstone, made up of the Alpujarride complex and Baetic/Penibaetic cordillera, covered with small flat stones.

Several webs have been observed in multiple similar biotopes in Andalusia. This singular web pattern turned out to be common, and adult and juvenile specimens both constructed it (including those who were kept alive in captivity). The very discreet webs are located under small stones on the ground. This structure provides the spider with protection against the elements: mainly intense heat but also wind and rare precipitation. Hunting canopies are very short and simple compared to those woven by species of *Eresus*. The details of a retreat are illustrated in Figure 10A–E. Sectional views of the canvas and lodge assembly (between the stone and the ground) are depicted in Figure 10A,B. The main lodge, located below the surface of the stone, is the main living space. Females have been observed in captivity to consume their prey and sometimes molt or copulate (sharing this lodge with the male) in this area. Hunting behavior is mainly sit-and-wait. Vibration is received from the external radial silk lines. The periphery of this lodge consists of dense cribellate silk, mixed with small pieces of agglomerated soil, anchored both to the ground and under the stone. From the main lodge, two separate exits with two capture canopies exist. The silk retreat is covered with a trapdoor-like hatch made of thick cribellate silk with soil particles in it (Figure 10C–E).

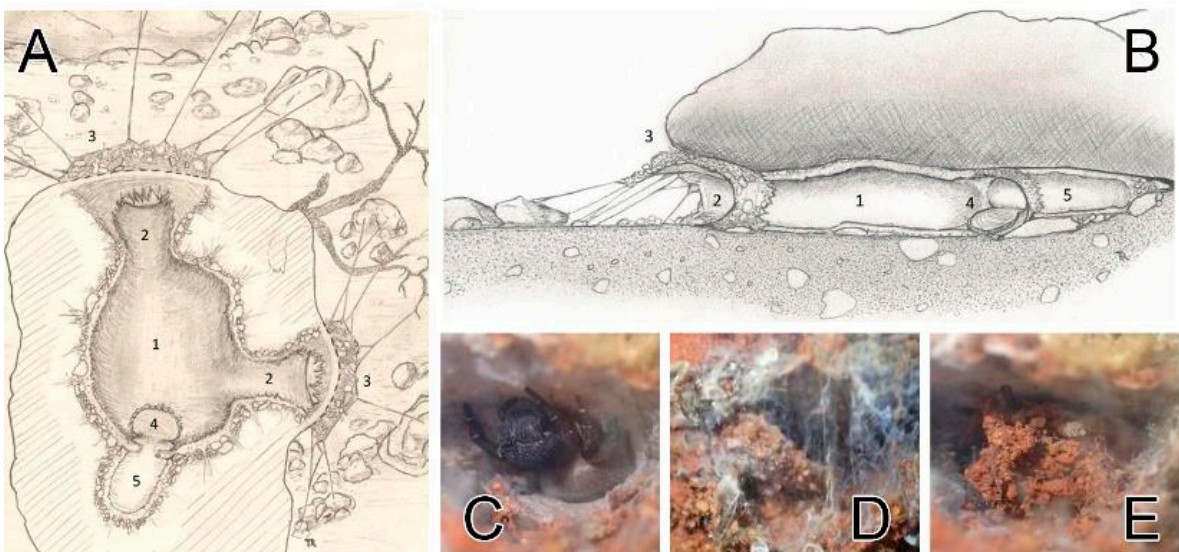

**Figure 10.** Retreats of *Loureedia colleni* from Spain. (**A**,**B**) retreat layout, dorsal and lateral views (illustration: Thomas Romanoff); (**C**) female with the open hatch; (**D**) hatch being closed; (**E**) closed hatch. Legend (**A**,**B**): **1**—main lodge; **2**—external access tunnels; **3**—hunting canopy and radial silk lines; **4**—"retreat" hatch; **5**—annex "retreat" lodge.

Phenology. Males are active during February–November.

Distribution. Spain (Albacete, Alicante, Almería, Ciudad Real, Granada, Madrid, and Murcia provinces) (see Figure 5).

*Loureedia jerbae* (El-Hennawy, 2005).

Figures 1D, 2D, 4B, 5, 6B, 7B, 8B and 11A,B.

*Eresus jerbae* El-Hennawy, 2005: 88, Figs 1–4 (♀) [11] (original description).

*Loureedia annulipes* Miller et al. 2012: 88 [1] (synonymy with *L. annulipes*; **rejected here**).

Type material. Holotype: female (MNHN 471/AR 835), TUNISIA: Djerba; misidentified as *Eresus petagnae* (not examined).

Other examined material. One male (HNHM), TUNISIA: Djerba, Djerba Midun, 33°48′36.2″ N, 11°02′38.3″ E, X. 2019 (leg. S. Macík).

Diagnosis. The male palp of *L. jerbae* (Figures 4B, 6B, 7B and 8B) is most similar to that of *L. phoenixi* (Figures 4A, 6A, 7A and 8A), as the arms of the conductor are almost the same length and bear pointed tips and the terminal portion of the prolateral arm curves retrolaterally (Figure 4A,B). The male palp of *L. jerbae* differs from that of *L. phoenixi*, in that the longer stem of the conductor bears only a slight curvature along its ectal margin (Figures 4B and 7B) vs. a shorter stem with an abrupt invagination on the ectal margin (Figures 4A and 7A), the retrolateral arm of the conductor is slightly longer than the prolateral one (Figure 4B) vs. both arms of the same length (Figure 4A), and the base of the prolateral arm of the conductor is wider (Figures 4B and 7B). The male coloration pattern of *L. jerbae* (Figures 1D and 2D) is similar to those of *L. maroccana* (Figures 1C and 2B) and *L. lucasi* (Fig 2C, Henriques et al. [3]: Fig 1d); it differs from both species by having numerous white spots and short stripes at the tips of the lateral branches of the median abdominal foliate pattern (Figures 1D and 2D) vs. no white spots (Fig 2B, Gál et al. [2]: Fig 1) or only a few very small spots (Fig 2C, Henriques et al. [3]: Fig 1d). It also differs from *L. lucasi* by having a reddish posterior part on the carapace (Figure 2D) vs. dark (Figure 2C). The female of *L. jerbae* differs from that of *L. lucasi* by its longer than wide epigynal windows (see El-Hennawy [11]: Figs 1–4) vs. round (see Henriques et al. [3]: Fig 1e,f).

Description. Male. Habitus as in Figures 1D, 2D and 11A,B. Total length: ca. 8.00. Carapace: 4.61 long and 3.61 wide. Abdomen: 4.09 long and 3.49 wide. Eye sizes and inter-eye distances: AME 0.12, PME 1.89, ALE 0.03, PLE 0.03, AME–AME 0.09, and ALE–AME 0.30. The carapace, sternum, labium, chelicerae, and maxillae dark brown. Carapace mostly

covered with long black setae and scattered short crimson and white scales. Scale patches of short red setae present mostly on the sides of the pars thoracica and the center of the pars cephalica, with two white spots next to the PLE. Legs covered with thin black hairs, with distinct regions of white hairs at the joints of all segments, forming distinct white annulation (Figure 2D). Abdomen with a crimson red longitudinal foliate pattern with white lines at its lateral extensions. The most anterior part of the median globular pattern with three lobes: white lateral lobes and a crimson red anterior lobe. Measurements of legs: I: 9.11 (3.04, 1.55, 1.86, 1.58, 1.06); II: 8.85 (2.76, 1.62, 1.72, 1.63, 1.11); III: 7.57 (2.63, 1.39, 1.50, 1.32, 0.71); IV: 10.2 (3.25, 1.60, 2.25, 2.03, 1.03).

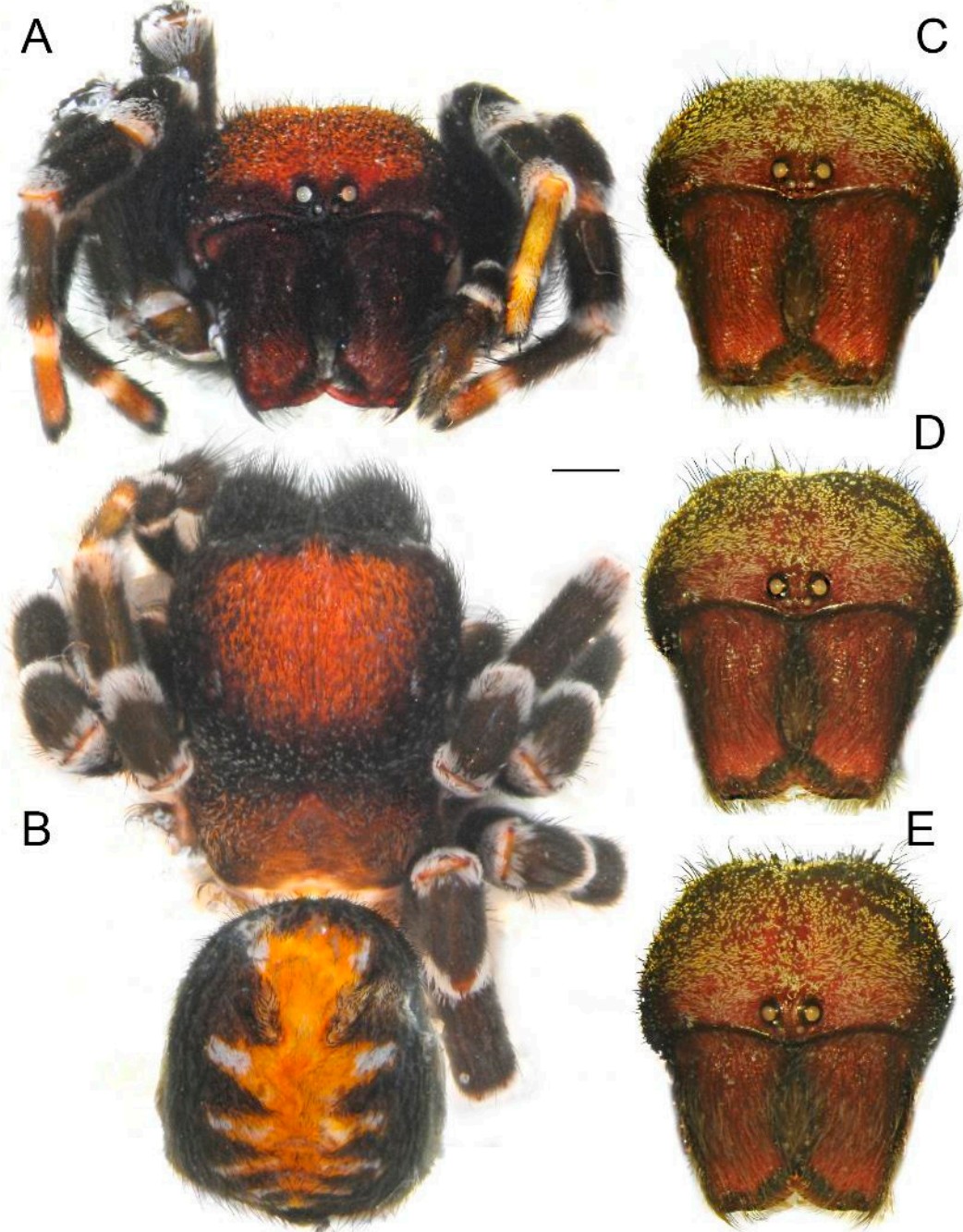

**Figure 11.** Frontal views and habitus of males of *Loureedia jerbae* and *Loureedia maroccana* (holotype). (**A**) *L. jerbae*, prosoma, frontal view; (**B**) *L. jerbae*, habitus, dorsal view; (**C**) *L. maroccana*, frontal view; (**D**,**E**) *L. maroccana*, slightly pushed down, showing changes in the perception of the carapace. Scale bar: 1.0 mm.

Palp as in Figures 4B, 6B, 7B and 8B. The stem of the conductor ca. two times longer than wide. The mesal margin of the conductor almost straight. The ectal margin with a slight medial invagination. The retrolateral arm of the conductor slightly longer than the prolateral arm. The retrolateral arm curves centrally, and both arms with pointed tips.

Female. See El-Hennawy [11], which is the only source regarding this species so far.

Variation. There are two observations of *Loureedia* from Tunisia on iNaturalist: one from Bizerte, with a very similar abdominal pattern to our specimen. The second specimen, although from Djerba, has noticeably larger white spots lateral to the median red band; two individuals with the same pattern have been photographed in northwestern Libya, not far from Djerba. Likely, these specimens belong to *L. jerbae*, although it is necessary to examine them to confirm this.

Natural history. No information.

Phenology. Males are active during October.

Distribution. Tunisia (Djerba) (see Figure 5).

*Loureedia maroccana* Gál, Kovács, Bagyó, Vári and Prazsák, 2017.

Figures 1B, 2C, 3, 4E, 6D,E, 7D,E, 8D,E, 11C–E, Figures 12–14.

*Loureedia maroccana* Gál et al., 2017: 12, Figs 1, 2, 3A–C and 4A–D (♂) [2] (original description).

*L. lucasi*: Henriques et al. 2018: 5 [3] (in part; synonymy with *L. lucasi*; **rejected here**).

Type material. Holotype: male (HNHM Araneae-8869), MOROCCO: Khémisset Province: near Sidi Boukhalkhal, 04.XI.2013 (leg. J. Gál). Paratypes: two males (HNHM Araneae-9007), same data as for the holotype (examined).

Other examined material. One male (PCGJ), MOROCCO: Khémisset Province: near Sidi Boukhalkhal, 04.XI.2013 (leg. J. Gál), and one male (PCGJ), MOROCCO: Khémisset Province: near Sidi Boukhalkhal, XI.2021 (leg. J. Gál).

Remark: For a closer examination, the right palp of the holotype was removed and illustrated. However, the retrolateral arm of the conductor was accidentally damaged by the first author (see Figure 12). One palp of one of the paratypes was removed for SEM, while the other one was also unintentionally damaged. These structures seemingly become fragile when deposited in 96% alcohol. Due to the relatively low number of specimens available, both the original (intact Figures 6E, 7E and 8E and 12A) and damaged states (Figures 6D, 7D, 8D and 12B,C) of the holotype's palp were illustrated.

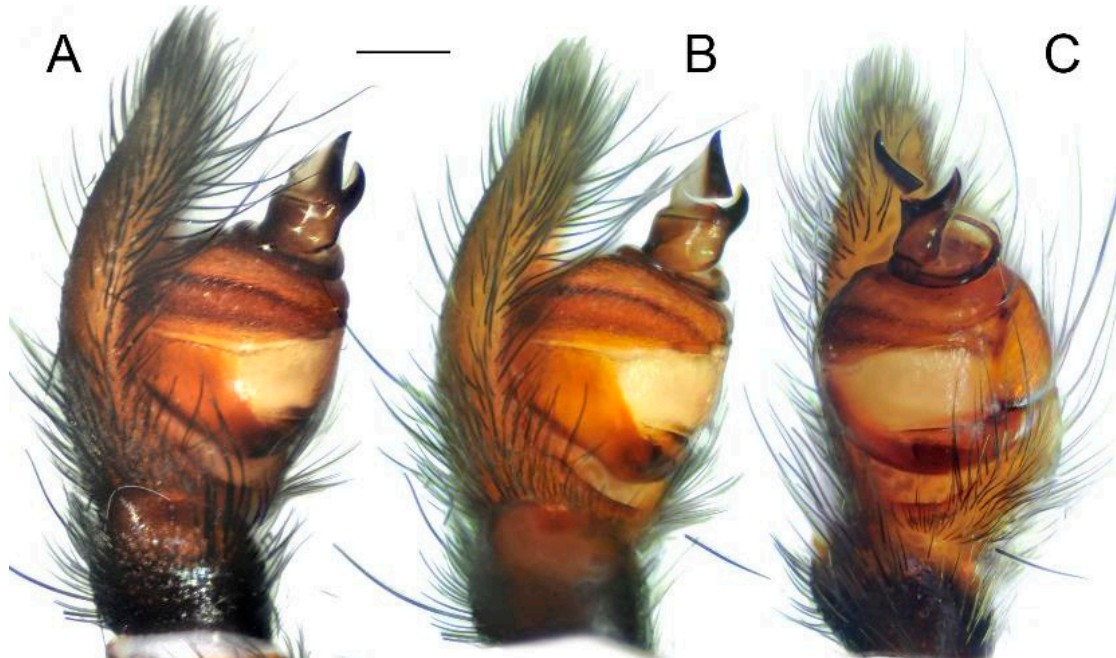

**Figure 12.** Holotype of *Loureedia maroccana* Gál et al., 2017 (damaged male palp). (**A**) Palp, before the damage, retrolateral view; (**B**) palp, after the damage (ra disconnected), retrolateral view; (**C**) palp, after the damage (ra disconnected), ventral view. Scale bar: 0.25 mm.

Diagnosis. The male palp of *L. maroccana* (Figures 3, 4E, 6E, 7E, 13 and 14A–N) is similar to that of *L. lucasi* (Figures 4D and 14O), as it has a relatively short stem of the conductor (Figure 4D,E) and the retrolateral arm of the conductor is longer than the prolateral arm (Figures 4D,E and 6E). It differs from it, as both arms of the conductor bear pointed tips (Figures 4E, 6D,E, 7D,E and 8D,E) vs. blunt (Figure 4D), there is a deeper concavity on the frontal margin of the conductor (Figures 3 and 4E), and there is an almost straight ectal margin (Figures 4E and 7D,E) at the stem of conductor (Figure 3A) vs. with a distinct invagination apically (Figure 4D). The basal margin of the conductor (Figure 3A) is also wider in *L. maroccana*, and the prolateral arm is wider and longer (Figures 4E and 7E) than that of *L. lucasi* (Figure 4D). The two species are also very similar in the male coloration pattern (Figure 2B,C) but differ due to the reddish posterior part of the carapace in *L. maroccana* (Figures 1C and 2B) vs. dark (Fig 2C, Henriques et al. [3]: Fig 1d). Moreover, the male of *L. maroccana* lacks minute white spots on the dorsal abdominal surface (Figure 1C) which are present in *L. lucasi* (Figure 2C).

Description. Male. Habitus as in Figures 1C, 2B and 10C–E. Total length: 8.77. Carapace: 4.39 long and 4.01 wide. Abdomen: 4.89 long and 3.97 wide. Eye sizes and inter-eye distances: AME 0.22, PME 0.19, ALE 0.12, PLE 0.20, AME–AME 0.39, and ALE–AME 1.08. The Carapace, sternum, labium, chelicerae, and maxillae dark brown with tones of red. Carapace mostly covered with long black and scarlet setae (these become orange (Figure 10C) when bleached in alcohol; see the specimen depicted in Gál et al. [2]: Fig 2B) and with very few scattered short white setae on the posterior part of pars cephalica. Pars cephalica well covered with scarlet red scales. Legs covered with thin black hairs, with distinct regions of white hairs at the joints, forming distinct white annulations. Abdomen with a compact longitudinal median red stripe with lateral projections with tiny white spots at their tips. The most anterior pair of the crimson red pattern quadrangle with three lobes. The lateral lobes with white tips. Leg measurements: I: 10.2 (3.21, 1.73, 1.95, 1.97, 1.33); II: 9.28 (3.05, 1.56, 1.72, 1.83, 1.10); III: 8.28 (2.78, 1.72, 1.55, 1.50, 0.71); IV: 10.4 (3.28, 1.66, 2.31, 2.15, 0.98).

Palp as in Figures 3, 4E, 6E, 7E, 8E, 12, 13 and 14. The stem of the conductor almost as long as wide (Figure 14). The mesal and ectal margins of the conductor with slight

curvatures (Figures 3 and 14). The retrolateral arm of the conductor ca. 1.5 times longer than the prolateral arm (Figure 13A–C) and slightly curved centrally. Both arms have pointed tips (Figures 4E and 13A–C).

Female. Unknown.

Natural history. No information.

Phenology. The males are active during October–November.

Distribution. Morocco (Khémisset Province) (see Figure 5).

*Loureedia phoenixi* Zamani and Marusik, 2020.
Figures 1F, 2F, 6, 7 and 8A.
*Loureedia* sp.: Henriques et al. 2018: 7, Fig 2h (♂) [3].
*Loureedia phoenixi* Zamani and Marusik, 2020: 240, Figs 1a–f, 2a–d and 3a–f (♂) [6] (original description).
*Loureedia phoenixi*: Zamani et al. 2021: 282, Fig 2A–D (♂) [22].

Type material. Holotype: male (MHNG), IRAN: Alborz Province: Karaj, Chenarak, 8.XI.2019 (leg. A. Beigi) (examined).

Other examined material. One male (ZMUT), IRAN: Tehran Province: Shemiranat County, Lavasan, 35°49′ N, 51°37′ E, 25.XI.2020 (leg. S. Bisadi).

Diagnosis. The male of *L. phoenixi* (Figures 4A, 6A, 7A and 8A) is similar to that of *L. jerbae* (Figures 4B, 6B, 7B and 8B), in the prolateral arm of the conductor being (almost) as long as the retrolateral arm. It can be readily distinguished from it by the shorter stem of the conductor (Figures 4A and 7A), and by the abdominal pattern, which is consisted of numerous large white spots on both the lateral and anterior margins of the median reddish band (Figures 1 and 2F).

Description. Male. Habitus as in Figures 1 and 2F. Total length: 5.55. Carapace: 2.72 long and 2.26 wide. Abdomen: 3.08 long and 2.12 wide. Eye sizes and inter-eye distances: AME 0.18, PME 0.15, ALE 0.03, PLE 0.04, AME–AME 0.28, and ALE–AME 0.57. Carapace, sternum, labium, chelicerae, and maxillae dark brown with tones of red. The carapace mostly covered with long black setae and scattered short white setae, with localized patches of short red setae, mostly in the pars thoracica or the center of the pars cephalica. Legs covered with thin black hairs, with distinct regions of white hairs at the joints of all segments, forming distinct white annulations. Abdomen with a compact longitudinal median red stripe with lateral projections with compact white spots at their tips. The most anterior pair of white spots either contiguous or very close to each other, sometimes merging and forming a distinct white spot above the pedicel. Measurements of legs: I: 6.70 (2.11, 1.02, 1.28, 1.40, 0.88); II: 5.94 (1.87, 0.98, 1.09, 1.24, 0.75); III: 4.89 (1.79, 0.68, 0.89, 0.96, 0.55); IV: 6.95 (2.19, 1.00, 1.61, 1.43, 0.69).

Palp as in Figures 4A, 6A, 7A and 8A. The stem of the conductor ca. 1.2 times longer than it wide. The mesal margin of the conductor with a slight curvature. The ectal margin with a distinct apical invagination. The prolateral and retrolateral arms of the conductor subequal in length, and both with pointed tips.

Female. Unknown.

Variation. The extent of the white abdominal patches is variable: the lateral patches may be connected to each other in a few specimens (Figure 1F), and the anterior patches are usually connected to each other and to the white plate above the pedicel (Fig 1a,d and f in Zamani and Marusik [6]), although exceptions have been recorded (Fig 1c,e in Zamani and Marusik [6]).

Natural history. Wandering males have primarily been collected in well-vegetated steppes but also in and around urban habitats (Figure 9C).

Phenology. The males are active during October–November.

Distribution. Iran (Alborz, Chaharmahal and Bakhtiari, Fars, Kerman, Qom, Semnan, Tehran, and Yazd provinces) (see Figure 5).

*Loureedia lucasi* (Simon, 1873).
Figures 2C, 4D, 5 and 13O.

*Eresus lucasi* Simon, 1873: 353, plate. 10, Figs 8 and 9 (♂♀) [8] (original description).
*Loureedia lucasi*: Henriques et al. 2018: 5, Fig 1a–h (♂♀) [3]; Zamani and Marusik 2020: 242, Fig 3h (♂) [6].

Type material. Syntypes: male and female (NMHN), ALGERIA: Oran (leg. H. Lucas) (not examined).

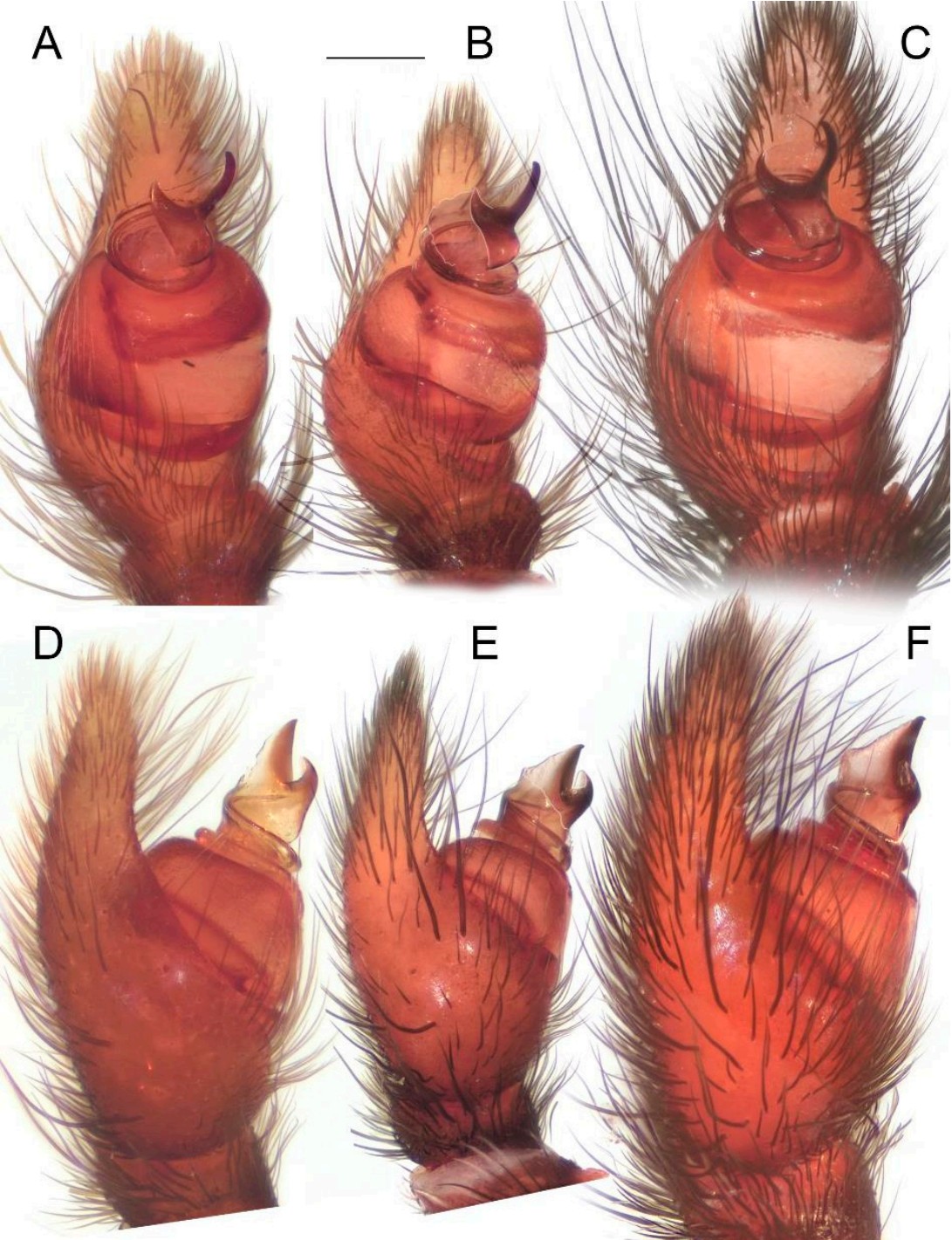

**Figure 13.** Intraspecific variation in the male palp of *Loureedia maroccana* Gál et al., 2017. (**A–C**) ventral view, (**D–F**) prolateral view. (**A,D**) non-type specimen; (**B,E**) paratype specimen; (**C,F**) fresh non-type specimen. Scale bar: 0.25 mm.

Diagnosis. The male palp of *L. lucasi* (Figure 4D) is similar to that of *L. maroccana* (Figure 4E) in that it has a relatively short stem of conductor and the retrolateral arm of

the conductor is longer than the prolateral arm, but differs from it, as both arms of the conductor bear blunt tips (Figure 4D) vs. pointed (Figure 4E), it has shallower concavity on the frontal margin of the conductor (Figure 4D), and there is a distinct invagination apically on the ectal margin of the stem of the conductor (Figure 4E). The two species are also very similar in the male coloration pattern but differ due to the dark posterior part of the carapace in *L. lucasi* (Figure 2C) vs. crimson red in *L. maroccana* (Figure 2B). Moreover, the male of *L. lucasi* has more distinct white spots on the dorsal abdominal surface (Figure 2C).

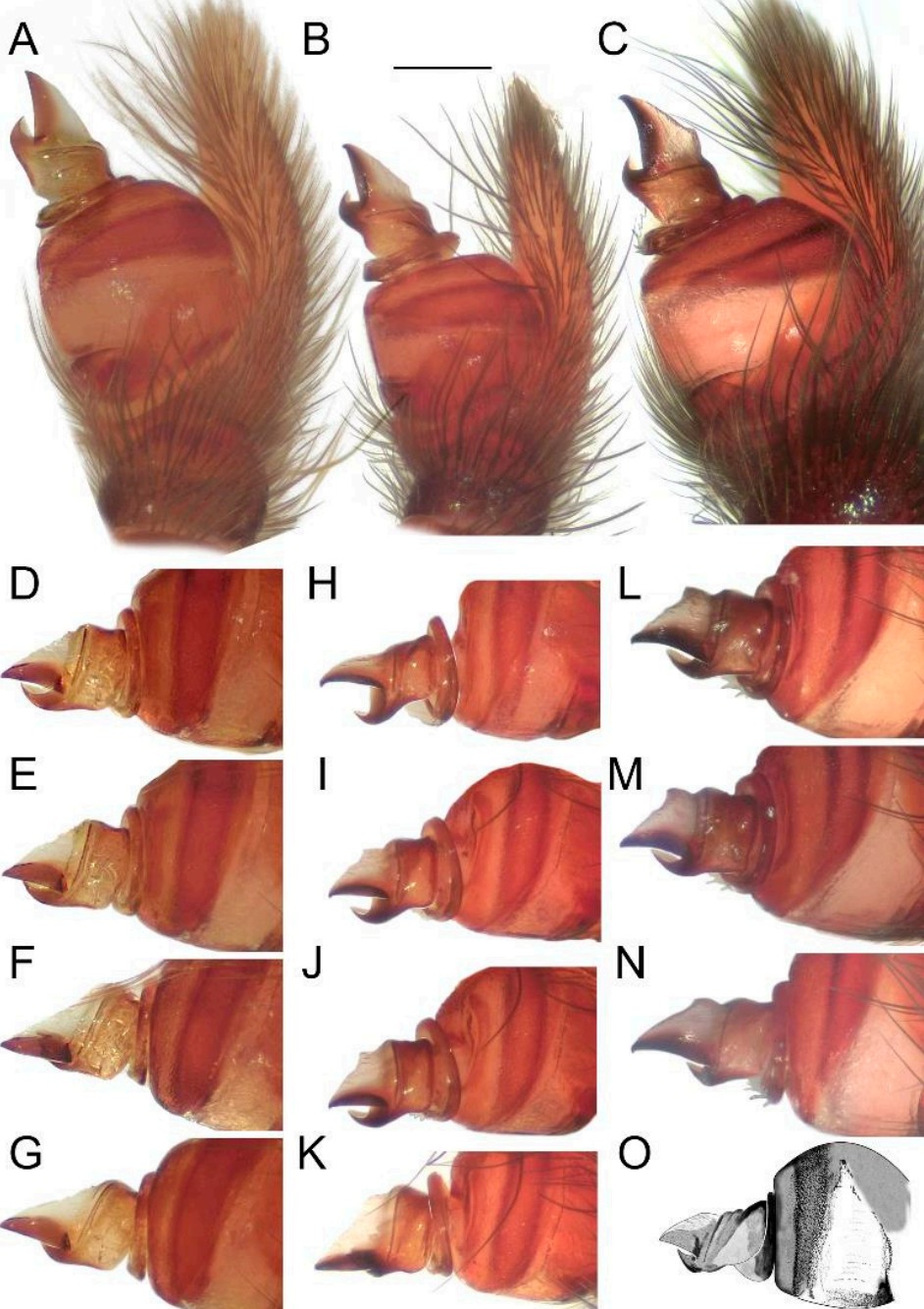

**Figure 14.** Effect of the angle on the perceived shape of the conductor in *Loureedia maroccana* Gál et al., 2017 (retrolateral views) (**A–N**). (**A,D–G**) non-type specimen; (**B,H–K**) paratype specimen; (**C,L–N**) fresh specimen; cymbial tip gradually turned in (**D–G,H–K,L–N**). (**O**) Male palp of *Loureedia lucasi* (Simon, 1873) (retrolateral view), redrawn based on Henriques et al. [3]. Scale bar: 0.25 mm.

Description. Henriques et al. [3] did not provide any description of the male, and the specimen was not available for us to examine.

Female. See Henriques et al. [3].

Variation. No information.

Phenology. No information.

Distribution. Algeria (Oran Province) (see Figure 5).

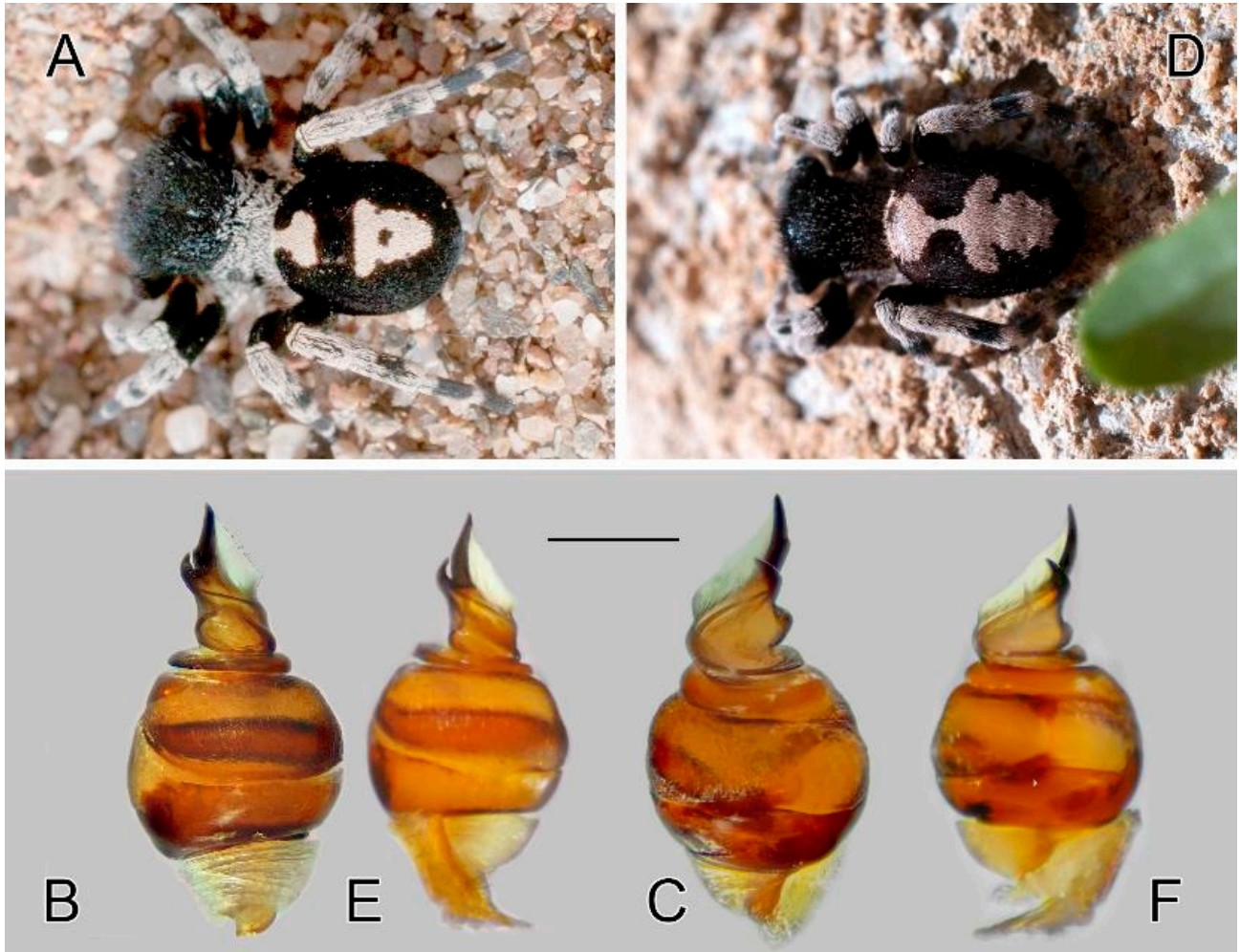

**Figure 15.** Intraspecific variation in *Loureedia colleni*. (**A–C**) male color form 1, (**D–F**) color form 2. (**A**) male habitus, dorsal view; (**B**) bulb, retrolateral view; (**C**) same, prolateral view; (**D**) male habitus, dorsal view; (**E**) bulb, retrolateral view; (**F**) same, prolateral view. Scale bar: 0.25 mm.

## 4. Discussion

The taxonomy (i.e., assigning specimens to properly named and adequately diagnosed taxonomic units and delimiting species [25]) of most eresid genera is notoriously difficult, which is perhaps best illustrated by the case of *Gandanameno* Lehtinen, 1967 in Miller et al. [1]. It was not possible to delimit the species in the genus, so it was necessary that "Gandanameno *species epithets removed to reflect increasing uncertainty about species limits and identity in this genus.*" The issue remains unsolved. In the description of *Eresus moravicus* Řezáč, 2008, a Central-European species, the author (i.e., the specialist who knew that species the best at the time) [26] wrote: "*We decided to designate material from only the southeastern part of the Czech Republic as paratypes, to minimize the possibility of including more taxa in the type*". As reported by Kraus and Kraus [27], both adult molting and color variation are unusually high, further complicating the precise assessment of species' delimiting characters in *Stegodyphus*. The only revised speciose genera remain to be *Seothyra* Purcell,

1903 [28] and *Stegodyphus* [27], although the monophyly of the latter has been questioned by a number of researchers (Král, pers. com.).

Traditionally spiders are primarily diagnosed and identified by their genitalia/copulatory organs [29]. In the case of *L. maroccana*, we examined all the available material, including the damaged holotype. Slight variation was observed in the curvatures of the prolateral arm and the retrolateral arm (Figure 13) of the conductor. As mentioned and documented by Zamani et al. [30] in *Eresus*, slight differences in the observation/imaging angle can greatly affect the perception of these structures (Figure 14). Therefore, regarding future studies on this group, we suggest that researchers include a number of full palp images from slightly varying angles. Including a full bulb-cymbium image (Figures 4, 6, 7, 8 and 14A–C) helps to orient the detailed structures and may also provide more information on, e.g., the extension of the medial haematodocha as a membranous triangle (Figure 14D–N), the curvature of the reservoir, and the ejaculatory duct, as noted by Henriques et al. [3]. Due to the relatively simple structures of the bulb [1,27], more readily diagnosable characters (e.g., genetic data, coloration, phenology, and habitat preferences) for Eresidae as a whole are needed. Considering the relative simplicity of the male palps in members of this family, the potential role of convergent evolution in the extreme similarity of palpal structures between otherwise geographically separated species cannot be ruled out. For example, the conductor of *L. annulipes* is extremely similar to that of *L. colleni*, despite the two species being geographically separated (i.e., Israel and Spain, respectively) and displaying contrastingly different color patterns (Figure 1B,C).

The color patterns in eresid males are peculiar. At least in *Eresus*, they are probably under selective pressure (ref. [31]; Pekár, pers. comm.) and are traditionally thought to be stable and diagnostic [3,32–34], although color variations were reported in many cases [3,27,30]. In *Loureedia*, only *L. colleni* has been collected in larger numbers [3], and male color forms are well documented. However, the copulatory structures of different variants have not been illustrated, and the conspecificity of these individuals has not been tested based on molecular methods. Indeed, we documented minor differences in the frontal margins of the conductors within the variants present in the small number of specimens that we studied (Figure 15), although these individuals are considered to be conspecific, following the results of the molecular analyses (see Figure 16), which was expected, as they were collected in the same area. This is in concordance with the case of *Gandanameno*, where the groupings were determined based on geography rather than morphology. However, this may also be an artefact due to the mitochondrial marker used in both cases, which is inherited only maternally, and females are much less mobile than males. This method only analyzes the resident female population structure, regardless of the number of incoming alleles. Because these are not showing up in the next generation, they remain invisible as they are not inherited from these males.

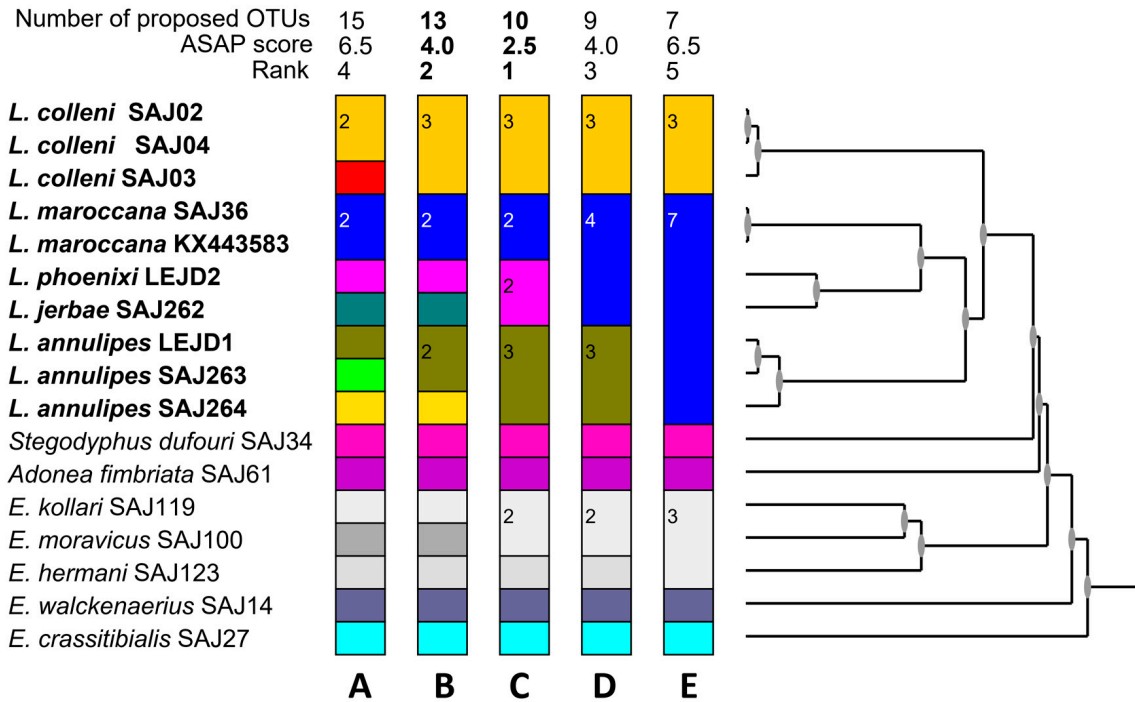

**Figure 16.** Result of the species delimitation analysis performed by ASAP. (**A**) 15 taxa grouping, with ASAP score 6.5; (**B**) 13 taxa grouping, with ASAP score 4.0; (**C**) 10 taxa grouping, with ASAP score 2.5; (**D**) 9 taxa grouping, with ASAP score 4.5; (**E**) 7 taxa grouping (*Loureedia* would have only two species) grouping, with ASAP score 6.5. Different colors correspond to recognized species.

Color pattern variation has also been observed in *L. maroccana* (see Gál et al. [2]: Figs 1 and 2A), and it is worth reporting that the scarlet pigment bleaches to orange within a relatively short time (after a decade in alcohol: Fig 10A–C, living specimen: Gál et al. [2]: Fig 2A). As noted regarding the palpal characters, the angle of the observation can also heavily influence the perception (Figure 14) of the somatic characters (e.g., the height of the carapace).

Unfortunately, we were not able to adequately examine the intraspecific variation in *L. annulipes*. Although slight variations were observed in the male abdominal patterns, i.e., the extension of the white patches, no differences were detected in the structures of the palp. In the ASAP analysis (see below), one specimen (SAJ264) was branched from the group of the other two identified as this species; unfortunately, this sample was included only based on an available DNA extract, and we were not able to study the actual specimen itself. Thus, we are unable to comment on the palp or the abdominal pattern.

Based on COI sequences, an ASAP analysis [16] was conducted as a preliminary approach (Figure 16). All species except for *L. lucasi* were included in the analysis: three sequences from *L. colleni* and *L. annulipes*, two from *L. maroccana*, and one from *L. jerbae* and *L. phoenixi*. The lowest ASAP score is shown in Figure 16C. In this grouping, *L. colleni*, *L. annulipes*, and *L. maroccana* are well separated, but oddly enough, *L. phoenixi* and *L. jerbae* are considered as the same species. This contrasts their geographic separation and the clear differences in the male color patterns, despite their palps being quite similar (Figure 4A,B). Furthermore, according to this grouping, *E. kollari* and *E. moravicus* are also considered conspecific, which contrasts the clear differences in the carapace shape, conductor shape, habitat, and phenology of these two species. As mentioned by Puillandre et al. [16] "*ASAP users should consider not only the partition with the best asap-score but also few subsequent ones*". Therefore, instead of simply using the lowest score, we compared the taxonomic knowledge (i.e., the separation of *E. kollari* from *E. moravicus* and *L. phoenixi* from *L. jerbae*) with the sequence similarities. Only two instances yielded these species as separate (Figure 16A,B), and we chose the lower ASAP score result. That was actually the second-best option

(Figure 16B) overall. Thus, we considered it as a working hypothesis. It recovered both the above-mentioned species pairs as separate taxa, but it also recovered a cryptic species in *L. annulipes*. However, a closer examination of the hierarchical cluster and the partitioning (see Figure 16B,C) shows a more detailed picture. Indeed, there is a structure in *L. annulipes* (as one would expect from such a slow dispersing spider species), but that difference is less in comparison to that between *L. jerbae* and *L. phoenixi*. Moreover, as mentioned by Collins and Cruickshank [34] "*coalescent depth among species vary considerably*". Thus, instead of fixed values to the distance thresholds, optimized thresholds need to be used [35,36]. Hence, we consider the previous grouping (in which the latter two (*L. jerbae* and *L. phoenixi*) are considered separate species; Figure 16B) to be more likely, at least until more data become available.

The preliminary Bayesian tree (Figure 17) offers only preliminary conclusions, although the informal grouping of Henriques et al. [3], i.e., all species but *L. colleni* grouped together, is not supported. In our analysis, *L. colleni* forms a clade with *L. annulipes* (although with low support of 0.82), whereas the monophyly of *L. maroccana*, *L. jerbae*, and *L. phoenixi* is well supported (Figure 17). However, taking into consideration that the tree is based on a single maternally inherited gene, we do not propose any further grouping within the genus until further, more thorough investigations are carried out.

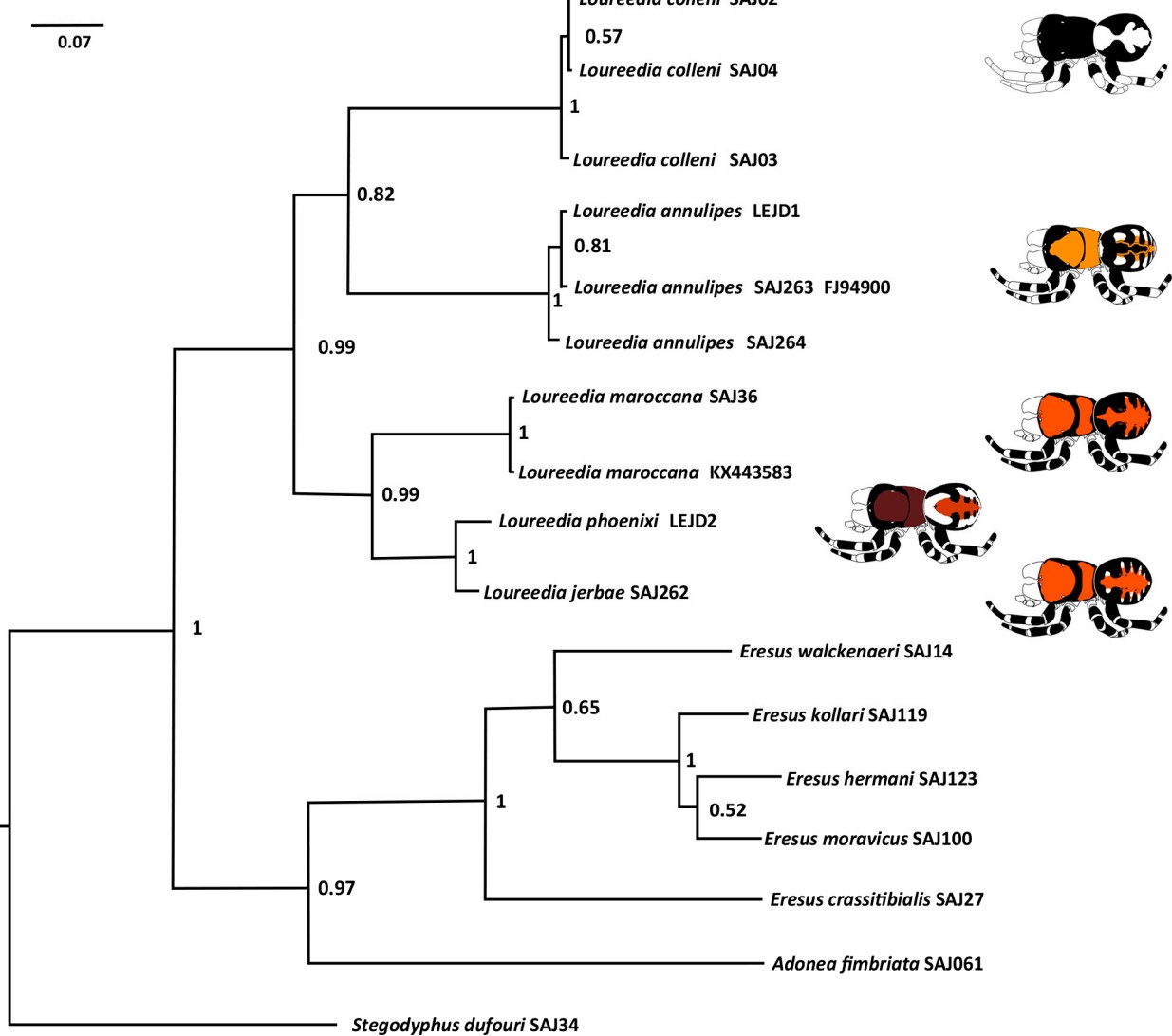

**Figure 17.** Bayesian phylogenetic tree based on COI sequences. Support values are posterior probability.

## 5. Conclusions

Our study has reviewed all the available *Loureedia* specimens and records. We illustrated and described five of the six known species. *Loureedia maroccana* was intensively illustrated, and based on observable differences we reinstated its name. Based on palpal and COI sequences, *L. jerbae* was also removed from its synonym with *L. annulipes*. All color patterns were illustrated. The effects of the color bleaching of specimens and the observational angle on the perception of the illustrated characters were shown. DNA barcoding suggested a cryptic species in *L. annulipes* and showed a closer split for *L. jerbae* and *L. phoenixi*. The preliminary phylogeny contradicted the previous grouping of the species (i.e., *L. colleni* separated from all the other species).

**Author Contributions:** Conceptualization, T.Sz. and A.Z.; methodology, K.Sz.; software, K.Sz.; validation, J.M. and M.F. (Martin Forman); formal analysis, K.Sz.; data curation, T.Sz.; writing—original draft preparation, T.Sz., A.Z., K.Sz. and J.M.; writing—review and editing, T.Sz., K.Sz., J.M., M.F. (Martin Forman) and M.F. (Magali Fabregat); visualization, T.Sz., A.Z., P.O., M.F. (Magali Fabregat) and G.K.; supervision, J.G. All authors have read and agreed to the published version of the manuscript.

**Funding:** Tamas Szűts was supported by a Naturalis Martin Fellowship for a 30-day stay in Leiden. Martin Forman was supported by the Ministry of Education, Youth, and Sports of the Czech Republic (projects SVV-260568 and LTAUSA19142).

**Institutional Review Board Statement:** Not applicable.

**Informed Consent Statement:** Not applicable.

**Data Availability Statement:** All data are freely available on GenBank.

**Acknowledgments:** We would like to thank everyone who helped this project in any way, including László Dányi (former curator), Eszter Lazányi-Bacsó (current curator) of HNHM, Assaf Usan, and Efrat Gavish-Regev of HUJ for lending material and allowing DNA extraction. We are indebted to Thomas Romanoff and Mahla Pourcheraghi for the drawings and to Grant McDonald for his help with the vector graphs as well as to Ersen Yağmur, Stanislav Macík, Csaba Szinetár, István Urák, Jørgen Lissner, Robert Bosmans, Johan Van Keer, Ed Nieuwenhuys, Reut Ein-Gil, Daniel Waysman, Amit and Einat Politi, and Itay Tessler for helping to obtain specimens, photos, and other relevant information. Three anonymous reviewers provided us with constructive criticism, which is highly appreciated.

**Conflicts of Interest:** The authors declare no conflict of interest.

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
