# Peer review of "A Study in Scarlet: Integrative Taxonomy of the Spider Genus Loureedia (Araneae: Eresidae)"

_diversity, doi:10.3390/d15020238_

Round 1

Reviewer 1 Report

I am delighted to read this revision of these beautiful spiders.  In southern Africa we have the velvet spiders Seothyra, which are like Loureedia in their markings and discrete distributions.  This Loureedia revision is very timely and solves several taxonomic problems.  I am also happy to see a discussion and illustration of their biology.

I recommend publication with only minor changes.  Specific lines in the text are referenced below.

2 – Please add “spiders” somewhere in the title.  The authors may want to change the title to “A Study in Scarlet: integrative taxonomy of the genus Loureedia (Araneae: Eresidae)” which commemorates the novel “A Study in Scarlet”, an 1887 detective novel by British writer Arthur Conan Doyle. The story marks the first appearance of Sherlock Holmes and Dr. Watson - Wikipediahttps://en.wikipedia.org › wiki › A_Study_in_Scarlet

or, the authors may want to contrast this title with one of their previous papers in velvet spiders:  - “The scarlet study: integrative taxonomy of the genus Loureedia (Araneae: Eresidae)” in contrast to “Zamani, A.; Altin, Ç.; Szűts, T. A Black Sheep in Eresus (Araneae: Eresidae): Taxonomic Notes on the Ladybird Spiders of Iran and Turkey, with a New Species. Zootaxa 2020, 4851, 559–572, doi:10.11646/zootaxa.4851.3.6.”

58 – “Fig. 1A L. annulipes from Israel, in defensive posture, with the bifid conductor visible” – the bifid conductor is very hard to see; maybe an arrow would clarify.

150, 151 –  “Simon, 1873, Loureedia and Dorceus C. L. Koch, 1846 [1] with Stegodyphus Simon, 1873 sister  to them is well established –  “well established” Please explain or cite  reference for this

180 – “Our voucher Loureedia specimens with additional data from Iran were also supplemented.” – Please explain how these data were supplemented, e.g., from unpublished records, from another pubication?

251 – “D L. maroccana, holotype (conductor broken); E L. maroccana, holotype (conductor intact)”  -- I was at first surprised to see what appeared to be two holotypes.   Are these before and after pictures?  Maybe clarify this, i.e., “D L. maroccana, holotype (after conductor broken); E L. maroccana, holotype (conductor still intact)” 

286, 287 – colleni H et al, 2018 “shorter white setae. White setae localized densely on pars thoracica and forming a triangle on pars cephalica” – I don’t see the white setae in Fig. 2A; perhaps not all specimens have these?

300, 301 – “intraspecific variations, as the COI sequences of the two males were more similar to each other, than to that of the female.” –  How much variation might exist in the species, and how does this compare to values that some use in an attempt to “barcode” species?

302 – Fig. 9.  This is a wonderful description of the biology of these spiders.  It would be even more informative to have a scale bar on the figures of the nest and nest entrance.

321 – change “lodge located below the surface of the stone, it is the main living space” to “lodge, located below the surface of the stone, is the main living space.”

327 –  change “are exist.” to “exist.”

376 – Loureedia should be in itlics

395-397 – “Other material examined. 1 male MOROCCO: Khémisset Province: near Sidi Boukhalkhal, 04.XI.2013 (leg. J. Gál). 1 male MOROCCO: Khémisset Province: near Sidi Boukhalkhal, XI.2021 (leg. J. Gál).”  Where are these specimens deposited?

469 – “ retrolateral and retrolateral arms” – I think that this should be “prolateral and retrolateral”

469 – “lenght” – change to “length”

474 – change “Za-mani” to ”Zamani“

505 – change ”was not giving” to” did not give”

517 – change ”delimit” to ”delimiting”

519 – change “so Gandanameno species epithets re-“ to “so it was necessary that Gandanameno species epithets be re-“

525 – change “color variation is in an unusually wide” to “color variation in an unusually wide”

527 – add space after “[26]”

558 – change “a larger” to “larger”

566 – change “rather morphology” to “rather than morphology”

570 – change “to report” to “reporting”

601 – change “i.e. all” to “i.e., all”

623 – change “extract DNA out” to “DNA extraction”

682 – reference 27 “Eberhard, W.G.; Huber, B.A. Spider Genitalia” is incomplete.

Author Response

We are very much indebted for this very friendly review. The first author personally is very happy that someone spotted the reference to the Doyle novel. Title modified to refer this novel properly.
2 – Please add “spiders” somewhere in the title. 

The title has been modified based on the reviewer's suggestions "spider" is added

58 – “Fig. 1A L. annulipes from Israel, in defensive posture, with the bifid conductor visible” – the bifid conductor is very hard to see; maybe an arrow would clarify.

Arrow added (legend modified, now it refers to it)

150, 151 –  “Simon, 1873, Loureedia and Dorceus C. L. Koch, 1846 [1] with Stegodyphus Simon, 1873 sister  to them is well established –  “well established” Please explain or cite  reference for this

Miller et al --> ref [1] added to clarify.

180 – “Our voucher Loureedia specimens with additional data from Iran were also supplemented.” – Please explain how these data were supplemented, e.g., from unpublished records, from another pubication?

We added the referring source

251 – “D L. maroccana, holotype (conductor broken); E L. maroccana, holotype (conductor intact)”  -- I was at first surprised to see what appeared to be two holotypes.   Are these before and after pictures?  Maybe clarify this, i.e., “D L. maroccana, holotype (after conductor broken); E L. maroccana, holotype (conductor still intact)” 

Changes applied, in all legends

286, 287 – colleni H et al, 2018 “shorter white setae. White setae localized densely on pars thoracica and forming a triangle on pars cephalica” – I don’t see the white setae in Fig. 2A; perhaps not all specimens have these?

Fig. 2A shows major patterns, the white setae are too delicate to show then on such drawing (they present in most species), hence it is ommited from there. However, they are clearyly visible on Fig 1E, which referenced here to clear the confusion.

300, 301 – “intraspecific variations, as the COI sequences of the two males were more similar to each other, than to that of the female.” –  How much variation might exist in the species, and how does this compare to values that some use in an attempt to “barcode” species?

2 males were identical, males - female 99,695% Unfortunately our data were not enough to make comments on the issue.

302 – Fig. 9.  This is a wonderful description of the biology of these spiders.  It would be even more informative to have a scale bar on the figures of the nest and nest entrance.

Unfortunately no scale bar was available, no measurements were recorded.

321 – change “lodge located below the surface of the stone, it is the main living space” to “lodge, located below the surface of the stone, is the main living space.”

Changed

327 –  change “are exist.” to “exist.”

Changed

376 – Loureedia should be in itlics

Changed

395-397 – “Other material examined. 1 male MOROCCO: Khémisset Province: near Sidi Boukhalkhal, 04.XI.2013 (leg. J. Gál). 1 male MOROCCO: Khémisset Province: near Sidi Boukhalkhal, XI.2021 (leg. J. Gál).”  Where are these specimens deposited?

in the private collection of the last author. PCJG - Private collection of Janos Gál, this also has been added to the specimen deposition information in the Materials section.

469 – “ retrolateral and retrolateral arms” – I think that this should be “prolateral and retrolateral”

changed

469 – “lenght” – change to “length”

changed

474 – change “Za-mani” to ”Zamani“

changed

505 – change ”was not giving” to” did not give”

changed

517 – change ”delimit” to ”delimiting”

changed

519 – change “so Gandanameno species epithets re-“ to “so it was necessary that Gandanameno species epithets be re-“

changed

525 – change “color variation is in an unusually wide” to “color variation in an unusually wide”

changed

527 – add space after “[26]”

added

558 – change “a larger” to “larger”

changed

566 – change “rather morphology” to “rather than morphology”

changed

570 – change “to report” to “reporting”

changed

601 – change “i.e. all” to “i.e., all”

changed

623 – change “extract DNA out” to “DNA extraction”

changed

682 – reference 27 “Eberhard, W.G.; Huber, B.A. Spider Genitalia” is incomplete.

completed

Reviewer 2 Report

This is a nice study summarizing current knowledge on an enigmatic spider genus, the velvet spider Loureedia. It is a useful taxonomic amendment. I have only minor issues, mostly grammatical, to point out. However, I think this should serve as a complete ´single stop´ review of the genus and thus urge the authors to describe both genders of all species rather than referring to prior publications for descriptions. Below is a list of minor edits that are needed to improve the manuscript.

´, hinting a´ Should read ´, hinting at a..’

I´m not familiar with the use of the word ´excitatory organ´ in spiders

Explain why you were not able to examine the holotype of E. lucasi

Define ´ASAP´ analysis before first use of abbreviation

It is unclear why default burn-in was used, rather than the point at which standard deviation of split frequencies reached thresholds

There is a gap between lines 213-214

Given that some species are known from fresh specimens (or in-field photography) and from alcohol storage, it would be useful to include in descriptions how coloration changes for each species in alcohol

´ Minor variations also occur on the male palp; these are considered 299intraspecific variations, as the COIsequences of the two males were more similar to each 300other,than to that of the female´

Here more detail would be useful. How similar were they? How certain are you regarding the match between males and female?

´underdeveloped´ is a poor choice of a word

´Female. SeeEl-Hennawy [11].´ Since the goal of this paper is to summarize current knowledge it would seem useful to include this information here so that the paper stands completely alone.

´but also in urban habitats and around urban areas´ awkward. Aren´t urban habitats around urban areas?

´Female. See Henriques et al.[3].´ See comment above.

Line 517 should be ´delimiting´ not ´delimit´

´adult  molting and color variation  is in an  unusually wide range further complicate precise  assessment of  species delimiting characters in Stegodyphus´ fix grammar e.g. color variation is unusually high further complicating…

Add space between ´ [26]and´

The legend for Figure 15 is incomplete. Please explain colors and details of subset and asap scores in legend. Further, there are two undistinguishable blue colors used (maroccana and crassitibialis) and it is also unclear which blu the dot in map indicates. Clarify with a more detailed legend and better choice of colors.

´more testable characters´ perhaps ´more readily diagnosable characters´?

Line 550 omit ´different´

´collected in a larger numbers´ fix grammar

´rather morphology´ should be ´rather than morphology´

´However, this may also be an artefact due to the mitochondrial marker used in both cases, which is inherited only maternally and females are much less mobile than males.´ Explain in more detail

Line 575 ‘were  observed  in  the  patterns´ which patterns?

´one specimen (SAJ264) was grouped separately from the other two identified as this species´ not according to phylogeny on Figure 16…

´habitus drawings are proportionate with the number of sequences available´ Seems like a strange thing to do. Make them proportionate with actual size and indicate the number of sequences by numbers.

´Figure 15 right´ be more precise 

´will became available´ should be ´will become available´ and remove the remainder of this sentence.

The conclusion section is incomplete. Please summarize the entire study.

Author Response

This is a nice study summarizing current knowledge on an enigmatic spider genus, the velvet spider Loureedia. It is a useful taxonomic amendment. I have only minor issues, mostly grammatical, to point out. However, I think this should serve as a complete ´single stop´ review of the genus and thus urge the authors to describe both genders of all species rather than referring to prior publications for descriptions. Below is a list of minor edits that are needed to improve the manuscript.

Many thanks for the constructive criticism. We and the manuscript benefited greatly from it.

´, hinting a´ Should read ´, hinting at a..’

changed

I´m not familiar with the use of the word ´excitatory organ´ in spiders

It was a google translate – for the almost exactly same French term, we left it there to imply the archaic descriptive language.

we changed it to “bulb” as it corresponds to that

Explain why you were not able to examine the holotype of E. lucasi

Two authors have tried to contact the museum at the beginning of the study (during the years), we have received no reply. Our last certain information regarding the whereabouts of the specimen is the publication of Henriques et al. 2018.

Define ´ASAP´ analysis before first use of abbreviation

Defined, paper cited

It is unclear why default burn-in was used, rather than the point at which standard deviation of split frequencies reached thresholds

We checked convergence in Tracer 1.7. to determine whether the runs with MrBayes default burn-in have reached convergence. Stationarity was reached in all BI runs quite early (prior to 1M generations) and therefore a conservative burn-in of 2,5 millions generations (the default 25%) was applicable. This procedure was added to the text.

There is a gap between lines 213-214

deleted

Given that some species are known from fresh specimens (or in-field photography) and from alcohol storage, it would be useful to include in descriptions how coloration changes for each species in alcohol

In line 423 we added an explanatory sentence, where Fig 10 C-E (bleached out 10 years old specimen) compared with its own fresh photos

´ Minor variations also occur on the male palp; these are considered 299intraspecific variations, as the COIsequences of the two males were more similar to each 300other,than to that of the female´ – Here more detail would be useful. How similar were they? How certain are you regarding the match between males and female?

The two males were identical (100%) female was 99.695% à since another rev also noted this we added it to the manuscript.

´underdeveloped´ is a poor choice of a word

Changed to ‘simple’

´Female. SeeEl-Hennawy [11].´ Since the goal of this paper is to summarize current knowledge it would seem useful to include this information here so that the paper stands completely alone.

We added ‘, which is the only source regarding this species so far’

´but also in urban habitats and around urban areas´ awkward. Aren´t urban habitats around urban areas?

We changed it in and around urban areas

´Female. See Henriques et al.[3].´ See comment above.

We added ‘, which is the only source regarding this species so far’

Line 517 should be ´delimiting´ not ´delimit´

changed

´adult  molting and color variation  is in an  unusually wide range further complicate precise  assessment of  species delimiting characters in Stegodyphus´ fix grammar e.g. color variation is unusually high further complicating…

Grammar fixed

Add space between ´ [26]and´

Added

The legend for Figure 15 is incomplete. Please explain colors and details of subset and asap scores in legend. Further, there are two undistinguishable blue colors used (maroccana and crassitibialis) and it is also unclear which blu the dot in map indicates. Clarify with a more detailed legend and better choice of colors.

Added and colors changed, explanation is given.

´more testable characters´ perhaps ´more readily diagnosable characters´?

changed

Line 550 omit ´different´

fixed

´collected in a larger numbers´ fix grammar

fixed

´rather morphology´ should be ´rather than morphology´

fixed

´However, this may also be an artefact due to the mitochondrial marker used in both cases, which is inherited only maternally and females are much less mobile than males.´ Explain in more detail

Explained in more detail

Line 575 ‘were  observed  in  the  patterns´ which patterns?

‘male abdominal patterns’ added

´one specimen (SAJ264) was grouped separately from the other two identified as this species´ not according to phylogeny on Figure 16…

Changed to branched, and added some explanation.

´habitus drawings are proportionate with the number of sequences available´ Seems like a strange thing to do. Make them proportionate with actual size and indicate the number of sequences by numbers.

We made them proportionate, the number of sequences indicated by the terminals.

´Figure 15 right´ be more precise 

We added letters to the columns to be more precise.

´will became available´ should be ´will become available´ and remove the remainder of this sentence.

Changed.

The conclusion section is incomplete. Please summarize the entire study.

Summarized the entire study, expanded this section considerably.

Reviewer 3 Report

This study performs an integrative revision of Loureedia, which clarifies the identities of some species that had been previously considered synonyms. The text is well written and the images and descriptions are good. I have only minor comments:

In all instances of Loureedia jerbae (El-Hennawy, 2005) the author must come between parentheses, as this is not the original generic combination. Please revise.

"Figure 4. Conductors of six species of Loureedia." Please indicate the view.

"Figure 5. D L. maroccana, holotype (conductor broken); E L. maroccana, holotype (conductor intact)" Which palp is which? Please indicate what figure has been mirrored.

Why have you not provided photos of L. lucasi? Is it possible to loan the original specimen?

"This is in concordance with the case of Gandanameno, where the groupings were determined by geography, rather morphology" > rather than

László Dányi (former curator) Eszter Lazányi-Bacsó (current curator) of HNHM), Assaf Usan, Efrat GavishRegev of (HUJ) > please revise parentheses

Author Response

I have only minor comments:

Many thanks for the review, whihc pointed out errors which we missed.

In all instances of Loureedia jerbae (El-Hennawy, 2005) the author must come between parentheses, as this is not the original generic combination. Please revise.

Revised, changed

"Figure 4. Conductors of six species of Loureedia." Please indicate the view.

it is ventral view, added.

"Figure 5. D L. maroccana, holotype (conductor broken); E L. maroccana, holotype (conductor intact)" Which palp is which? Please indicate what figure has been mirrored.

None of the figures has been mirrored. As pointed out by another reviewer, it was not explained adequately.
The only palp removed is illustrated, but it broke during the study. It is explained, and have been pőlaced in the figure legends.

Why have you not provided photos of L. lucasi? Is it possible to loan the original specimen?

We have not been successful to loan that specimen in the past 6 years.

"This is in concordance with the case of Gandanameno, where the groupings were determined by geography, rather morphology" > rather than

changed

László Dányi (former curator) Eszter Lazányi-Bacsó (current curator) of HNHM), Assaf Usan, Efrat GavishRegev of (HUJ) > please revise parentheses

revised, deleted the unnecessary ones.